# LAST: Bridging Vision-Language and Action Manifolds via Gromov-Wasserstein Alignment

**Huaihai Lyu** [1]   **Chaofan Chen** [1]   **Yuheng Ji** [1]   **Xiansheng Chen** [2]   **Pengwei Wang** [2]   **Shanghang Zhang** [3]
**Changsheng Xu** [1]

## Abstract

We take a Gromov-Wasserstein perspective on Vision-Language-Action (VLA) learning, where the goal is to make the relational geometry of action representations compatible with the semantic geometry of VL embeddings. However, this alignment is non-trivial due to the mathematical heterogeneity between the domains: the semantic space of vision-language is topologically linear and isotropic, whereas the physical manifold of robotic action is non-Euclidean and anisotropic. Their disjoint metric structures render direct regression ill-posed. To resolve this incompatibility, we introduce **LAST** (**L**ie-algebraic **A**ction **S**pace **T**okenizer), which reconstructs the action space to establish local metric compatibility with the VL modality via a two-stage transformation: (1) *Global Topological Linearization*: linearizing the action manifold via Lie-algebraic mapping, converting trajectories into a fixed-length, physically additive representation. (2) *Local Metric Discretization*: hierarchically discretizing the representation into schemas and whitened residuals, yielding approximately isotropic local charts that are statistically aligned with the semantic metric. By resolving the structural mismatch at both global and local levels, **LAST** enables VLA models with superior convergence and generalizability.

## 1. Introduction

The pursuit of generalist embodied intelligence has been accelerated by the rise of VLA models (Ji et al., 2025; Black

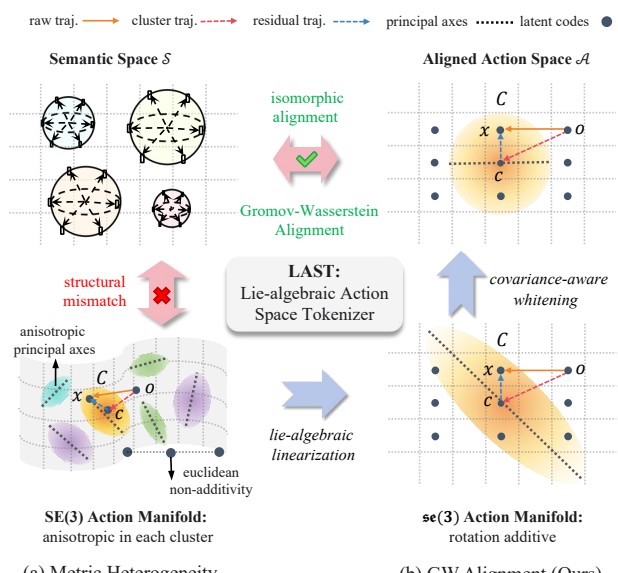

*Figure 1.* **Motivation of LAST.** (a) Semantic embeddings exhibit near-isotropic local neighborhoods under normalized cosine geometry (circles), whereas action modes are anisotropic (ellipses) and $SE(3)$ composition is non-additive. (b) **LAST** maps actions to the tangent space $\mathfrak{se}(3)$ for locally additive residuals, then performs covariance-aware whitening to reduce anisotropy.

et al., 2024). By treating robot actions as tokens, these models aim to inherit the scaling laws of Transformers (Sorscher et al., 2022). However, a fundamental theoretical barrier persists: *structural heterogeneity across VL and A modalities* (Rioux et al., 2024). To bridge this gap, we adopt a view of Gromov-Wasserstein (GW) alignment (Xu et al., 2019; Peyré & Cuturi, 2019) that compares relational geometry within each modality rather than pointwise coordinates.

Prior works (Saveriano et al., 2023; Florence et al., 2022) indicate that complex behaviors decompose into a small set of recurring primitives, yielding a multi-cluster structure (*e.g.*, the elliptical clusters in Figure 1(a)). When viewed in local coordinates, each elliptical cluster exhibits task-aligned anisotropy, where the principal axis concentrates variation along feasible dynamical directions (Paraschos et al., 2013). We therefore treat each cluster as a *prototypical action pattern*: $o$ is the coordinate origin, $c$ is a representative prototype centroid, and $x$ is an exact execution

[1]MAIS, Institute of Automation, Chinese Academy of Sciences.
[2]Beijing Academy of Artificial Intelligence. [3]Peking University.. Correspondence to: Chaofan Chen <chencfbupt@gmail.com>, Shanghang Zhang <shanghang@pku.edu.cn>.

*Proceedings of the $43^{rd}$ International Conference on Machine Learning*, Seoul, South Korea. PMLR 306, 2026. Copyright 2026 by the author(s).

trajectory in cluster $C$. This organization exposes a metric incompatibility between semantic embeddings and physical actions, resulting in two gaps that hinder cross-modal alignment. ***(1) Geometric Mismatch.*** Standard regression measures prototype deviation with a Euclidean residual, $\|\vec{ox} - \vec{oc}\|_2^2$, implicitly treating the action space as a convex vector space where additive updates remain feasible (*i.e.*, $\vec{ox} \approx \vec{oc} + \vec{cx}$). However, robotic actions lie on the Lie group $SE(3)$, which is non-convex and not closed under Euclidean addition (Barfoot, 2024). Consequently, Euclidean residuals and linear mixing can produce geometrically invalid rotations and mismeasure physical discrepancy (Zhou et al., 2019), causing off-manifold mixing and cross-modal misalignment. ***(2) Statistical Mismatch.*** VLM embeddings are trained contrastively and compared via cosine similarity on $\ell_2$-normalized features (Radford et al., 2021), encouraging a locally isotropic hyperspherical geometry (Wang & Isola, 2020). In contrast, action patterns are strongly anisotropic along dynamics-constrained directions (Sola et al., 2018). As a result, using the isotropic VL metric to retrieve action neighbors selects false neighbors across the ellipse's short axis while missing true neighbors along the principal axis. This neighbor drift breaks local metric consistency and destabilizes training.

To address these challenges, we take a Gromov-Wasserstein perspective on VLA learning, where the goal is to make the relational geometry of action representations compatible with the semantic geometry of VL embeddings. This GW-inspired structural alignment matches the intra-domain geometries rather than enforcing pointwise correspondence. To realize this view, we present **LAST** (**L**ie-algebraic **A**ction **S**pace **T**okenizer), which encodes manifold-valued actions to facilitate alignment with the semantic space. This structural alignment is realized through a dual-stage transformation: (1) **Global Topological Linearization**: We linearize the non-Euclidean action manifold by mapping trajectories into the Lie-algebraic tangent space $\mathfrak{se}(3)$ and parameterizing them via B-splines. This converts variable-length, curved trajectories into a fixed-length, physically additive vector representation, resolving the Euclidean non-additivity shown in Figure 1(a). (2) **Local Metric Discretization**: We hierarchically decompose the representation into discrete topological schemas and fine-grained residuals. Crucially, we employ covariance-aware whitening to rectify the statistical mismatch. As depicted in Figure 1(b), this transformation regularizes the skewed residual distributions toward approximately isotropic local charts, yielding an action space whose local geometry is statistically compatible with that of vision-language embeddings.

Building upon this foundation, we integrate **LAST** into a unified VLA training framework to enforce structural alignment between the semantic and action spaces. We first pre-train **LAST** to encode continuous trajectories into a compact discrete space via Global Topological Linearization, then apply Local Metric Discretization to hierarchically cluster motion patterns into discrete action tokens. During VLA training, we use two complementary supervisions. The backbone's hidden states are supervised in two ways: an AR head predicts **LAST** quantized tokens, shaping structured action intents, while a diffusion head is conditioned on the same hidden features to match continuous actions. Together, AR provides a stable discrete intent and diffusion refines continuous control details, improving control precision and generalization.

In summary, our contributions are threefold:

- We adopt a Gromov-Wasserstein perspective on action tokenization, identifying metric heterogeneity as the root cause of poor VLA generalization.
- We introduce **LAST**, a tokenizer that establishes local metric compatibility between action and semantic spaces via Lie-algebraic linearization and statistical whitening.
- Experiments on diverse simulation and real-world benchmarks demonstrate strong performance and generalization.

## 2. Related Work

### 2.1. Vision-Language-Action Models and Tokenization

Modern VLA policies (Kim et al., 2024; O'Neill et al., 2024) typically discretize actions to leverage the scalability of autoregressive Transformers. However, current tokenization often lacks geometric and statistical validity (Zhou et al., 2019). Simple binning (Kim et al., 2024) ignores the non-linear $SE(3)$ manifold, inducing kinematic distortions. Structured VQ-based methods (Lee et al., 2024) rely on Euclidean metrics that assume isotropic distributions, creating a statistical mismatch with the highly anisotropic nature of robotic motion. Furthermore, frequency-domain transforms like FAST (Pertsch et al., 2025) can introduce non-physical artifacts (*e.g.,* Gibbs ringing). These geometry-agnostic representations force models to overfit to artifacts rather than robust physical priors, limiting VLA generalization.

### 2.2. Geometric Alignment and Manifold Learning

Our work is grounded in geometric deep learning (Bronstein et al., 2021) and the theoretical analysis of multi-modal alignment (Liang et al., 2022). Recent studies (Mistretta et al., 2025) have identified a fundamental "Modality Gap" between visual-semantic embeddings and other heterogeneous geometric spaces. Direct regression between such disparate manifolds is often mathematically ill-posed without explicit structural alignment (Xu et al., 2019), frequently requiring Optimal Transport techniques to match intra-domain

relational geometries. Moreover, generative models on non-Euclidean data suffer from "Latent Space Oddity" (Arvanitidis et al., 2017), where the metric mismatch between flat priors and curved manifolds leads to severe distortion.

## 3. Methodology

### 3.1. Preliminaries

**VLA Learning Paradigm.** A VLA model learns a policy $\pi_\theta$ that generates action signals conditioned on visual observations $\boldsymbol{o}$ and language instructions $\boldsymbol{l}$. For notation simplicity, we introduce state $\boldsymbol{s} = [\boldsymbol{o}, \boldsymbol{l}]$. To capture temporal dependencies, the prediction target is typically an action chunk $\boldsymbol{a} \in \mathbb{R}^{T \times d}$, where $T$ denotes the action horizon and $d$ is the action dimensionality. The VLA is trained to minimize the negative log-likelihood:

$$\min_\theta \mathbb{E}_{(\boldsymbol{a}, \boldsymbol{s}) \sim \mathcal{D}} \big[ -\log \pi_\theta(\boldsymbol{a} \mid \boldsymbol{s}) \big]. \tag{1}$$

**Residual Vector Quantization.** Since standard Auto-Regressive (AR) backbones operate on discrete tokens, the continuous action chunk $\boldsymbol{a}$ must be discretized. To achieve high-fidelity compression, RVQ (Lee et al., 2024) is a standard approach. It approximates a target vector $\boldsymbol{z}$ through a coarse-to-fine summation of $L$ discrete codewords:

$$\hat{\boldsymbol{z}} = \text{RVQ}(\boldsymbol{z}) = \sum_{l=1}^{L} \boldsymbol{c}_{k_l}^{(l)}, \quad \text{where } \boldsymbol{c}^{(l)} \in \mathcal{C}^{(l)}. \tag{2}$$

Here, $\mathcal{C}^{(l)}$ denotes the codebook at layer $l$, and $\boldsymbol{c}_{k_l}^{(l)}$ is the codebook vector retrieved from $\mathcal{C}^{(l)}$ at index $k_l$.

### 3.2. Theoretical Analysis: A Gromov-Wasserstein Perspective

We take a Gromov-Wasserstein (GW) perspective on generalist VLA policy learning, viewing the policy as inducing a mapping $\Phi : \mathcal{X} \to \mathcal{Y}$ between two heterogeneous metric-measure spaces (Xu et al., 2019). The semantic space $\mathcal{X} = (\mathbb{R}^{d_s}, d_\mathcal{X}, \mu)$ comprises feature vectors in $\mathbb{R}^{d_s}$ distributed according to probability measure $\mu$ (*i.e.,* data distribution of VL embeddings), equipped with the Euclidean metric $d_\mathcal{X}$ arising from the statistically isotropic tensor $G_\mathcal{X} \approx I$. The action manifold $\mathcal{Y} = (\mathcal{M}, d_\mathcal{Y}, \nu)$ represents the distribution $\nu$ of valid trajectories on the submanifold $\mathcal{M} \subset SE(3)^T$, where $T$ denotes the action horizon. The intrinsic geodesic metric $d_\mathcal{Y}$ is governed by the highly anisotropic Riemannian tensor $G_\mathcal{Y} \not\approx I$. Rather than explicitly optimizing a GW objective, we leverage this perspective to design a tokenizer that makes the intra-domain relational geometries of $\mathcal{X}$ and $\mathcal{Y}$ structurally compatible.

Standard approaches typically employ direct regression to approximate $\Phi$. However, it is ill-posed due to (i) *geometric mismatch:* assuming a closed, additive Euclidean action space while $\mathcal{Y}$ is multi-cluster and non-additive; and (ii) *statistical mismatch:* assuming isotropy while $\mathcal{Y}$ is highly anisotropic, leading to ill-conditioned and unstable optimization. We formalize these failure modes in Lemma A.1 and Lemma A.2. Motivated by this analysis, we elaborate on how our design rectifies the action manifold.

#### 3.2.1. MANIFOLD RECTIFICATION

To achieve structural alignment, we rectify the mismatched action manifold along two axes. For geometric alignment, we first recover approximate intra-cluster additivity by linearizing actions in Lie-algebra tangent charts (see Lemma A.4 for details due to page limit). We then enforce local closure on the multi-cluster manifold by introducing discrete action intents (Lemma 3.1). For statistical alignment, we normalize local anisotropy via covariance-aware whitening to improve conditioning and stabilize GW alignment (Lemma 3.2).

**Lemma 3.1** (Local Closure via Discrete Action Intents). *Assume a multi-cluster action distribution modeled as a Gaussian mixture $P(\boldsymbol{a}|\boldsymbol{s}) = \sum_{k=1}^{K} \omega_k(\boldsymbol{s}) \mathcal{N}(\boldsymbol{a}; \boldsymbol{\mu}_k, \boldsymbol{\Sigma}_k)$, where the component means $\boldsymbol{\mu}_k \in \mathcal{M}$ represent distinct feasible action prototypes. Introducing a discrete intent variable $z \in \mathcal{Z} = \{z_k\}_{k=1}^{K}$ yields the locally closed factorization $P(\boldsymbol{a}|\boldsymbol{s}) = \sum_{k=1}^{K} P(\boldsymbol{a}|z = k, \boldsymbol{s}) P(z = k|\boldsymbol{s})$, which restricts each conditional transport to a corresponding feasible action chart in $\mathcal{Y}$.*

**Proof.** By introducing the discrete variable $z$, we factorize the joint distribution via the chain rule:

$$P(\boldsymbol{a}, z|\boldsymbol{s}) = P(\boldsymbol{a}|z, \boldsymbol{s}) P(z|\boldsymbol{s}). \tag{3}$$

This decomposes the optimization into two tractable subproblems: *(i) Mode Selection:* The term $P(z = k|\boldsymbol{s})$ learns to approximate the mixing weights $\omega_k(\boldsymbol{s})$, which is a discrete classification problem with a convex cross-entropy loss. This maps $\boldsymbol{s}$ to the correct Voronoi cell $\mathcal{X}_k$ (the region assigned to $z = k$). *(ii) Mode Refinement:* Conditioned on a specific token $z = k$, the target distribution becomes:

$$P(\boldsymbol{a}|z = k, \boldsymbol{s}) \approx \mathcal{N}(\boldsymbol{a}; \boldsymbol{\mu}_k, \boldsymbol{\Sigma}_k). \tag{4}$$

The corresponding energy function becomes *locally convex:*

$$\mathcal{E}_k(\boldsymbol{a}) = -\log P(\boldsymbol{a}|z = k) \propto \|\boldsymbol{a} - \boldsymbol{\mu}_k\|_{\boldsymbol{\Sigma}_k^{-1}}^2. \tag{5}$$

Thus, the global transport plan $\Phi$ decomposes into a mixture of local plans: $\Phi \approx \sum_k \Phi_k$, where each local plan $\Phi_k$ maps a semantic cluster $\mathcal{X}_k$ to a geodesically convex action chart $\mathcal{Y}_k$, ensuring consistency within each domain. Consequently, the structural alignment problem decomposes into a mixture of low-distortion local couplings, significantly easing the cross-modal matching. $\square$

**Lemma 3.2** (Statistical Alignment via Whitened Isomorphism). *Given actions in chart $z_k$ with local statistics $(\boldsymbol{\mu}_k, \boldsymbol{\Sigma}_k)$, applying a Covariance-Aware Whitening transformation $\psi_k(\boldsymbol{a}) = \boldsymbol{\Sigma}_k^{-1/2}(\boldsymbol{a} - \boldsymbol{\mu}_k)$ within a local chart $z_k$ aligns the anisotropic physical metric $G_{\mathcal{Y}}$ with the isotropic semantic metric $G_{\mathcal{X}}$, satisfying the Bi-Lipschitz continuity condition for the decoding function.*

**Proof.** The optimization difficulty is governed by the curvature of the loss landscape, represented by the Hessian matrix $H$. In the original space, the local metric tensor is $G_{\mathcal{Y}} \approx \boldsymbol{\Sigma}_k^{-1}$, leading to a Hessian condition number $\kappa(H) \approx \lambda_{\max}(\boldsymbol{\Sigma}_k)/\lambda_{\min}(\boldsymbol{\Sigma}_k) \gg 1$. The whitening transformation $\psi_k(\boldsymbol{a}) = \boldsymbol{\Sigma}_k^{-1/2}(\boldsymbol{a} - \boldsymbol{\mu}_k)$ acts as a preconditioning operator on the manifold geometry. Under this change of basis, the effective metric tensor transforms as:

$$\tilde{G}_{\mathcal{Y}} = (\boldsymbol{\Sigma}_k^{1/2})^{\top} G_{\mathcal{Y}}(\boldsymbol{\Sigma}_k^{1/2}) = \boldsymbol{\Sigma}_k^{1/2}\boldsymbol{\Sigma}_k^{-1}\boldsymbol{\Sigma}_k^{1/2} = I. \quad (6)$$

Consequently, the target distribution becomes isotropic, i.e., the whitened action distribution is approximately $\mathcal{N}(0, I)$. The mapping task reduces to finding a transport map between two isotropic spaces $\mathcal{X} \to \psi_k(\mathcal{Y}_k)$. The Jacobian $\mathbf{J}_{\Phi}$ of such an optimal map is approximately orthogonal, satisfying $\mathbf{J}_{\Phi}\mathbf{J}_{\Phi}^{\top} \propto I$. This implies the condition number collapses to unity ($\kappa(\mathbf{J}_{\Phi}) \approx 1$), directly satisfying the ideal **Bi-Lipschitz** constraint with constant $C \approx 1$:

$$\|\psi_k(\Phi(x_1)) - \psi_k(\Phi(x_2))\|_2 \approx \|x_1 - x_2\|_2. \quad (7)$$

This statistical rectification guarantees that the gradient descent steps are isotropic, ensuring more reliable GW alignment across modalities. $\qquad\square$

### 3.3. LAST: Lie-Algebraic Action Space Tokenizer

Based on the above analysis, we propose **LAST**, a Lie-algebraic action space tokenization framework instantiating the GW perspective via structural alignment of intra-domain geometries. We structurally decompose the process into two phases: (1) Global Topological Linearization: transforming the curved, variable-length manifold into a unified, fixed-length linear vector space; and (2) Local Metric Discretization: partitioning this linear space into local schemas and rectifying the internal geometry of each partition for optimal quantization. The complete pipeline is illustrated in Fig. 2.

#### 3.3.1. GLOBAL TOPOLOGICAL LINEARIZATION

To resolve the non-additivity issue, the first objective is to map the non-Euclidean action manifold into a globally consistent linear space. This produces a unified linear vector space where arithmetic operations are physically valid.

**Lie-Algebraic Linearization.** To ensure physical additivity, we map the raw trajectory $\boldsymbol{a} = \{P_t\}_{t=1}^{T} \in SE(3)^T$ into the Lie-algebraic tangent space, where $T$ denotes the action horizon. We compute the relative motion with respect to the start frame $P_1$ and apply the logarithmic map to obtain tangent vectors $\boldsymbol{\xi}_t \in \mathbb{R}^6$:

$$\boldsymbol{\xi}_t = \log(P_1^{-1} \cdot P_t)^{\vee}. \quad (8)$$

Here, the logarithmic map projects the manifold element onto the Lie algebra $\mathfrak{se}(3)$, a space of $4 \times 4$ matrices characterized by skew-symmetric rotation blocks. The operator $(\cdot)^{\vee}$ denotes the isomorphism that maps this matrix algebra to the Euclidean coordinate vector $\mathbb{R}^6$, where motion composition is approximately additive within a local chart (shown in Lemma A.4). Consequently, the action chunk is formed by stacking these vectors: $\boldsymbol{\xi} = [\boldsymbol{\xi}_1, \ldots, \boldsymbol{\xi}_T]^{\top}$. This transforms the curved manifold into a linear vector space, ensuring that subsequent linear operations preserve geometric validity.

**B-Spline Temporal Abstraction.** To handle temporal redundancy and variable horizon lengths, we parameterize the sequence $\boldsymbol{\xi}$ using B-splines. We solve for a sparse set of control points $\boldsymbol{z} \in \mathbb{R}^{T_c \times 6}$, where $T_c$ is a fixed latent horizon. We use ridge regression to suppress high-frequency teleoperation noise and prevent overfitting:

$$\boldsymbol{z} = \arg\min_{\boldsymbol{z}'} \|\boldsymbol{\Phi}\boldsymbol{z}' - \boldsymbol{\xi}\|_F^2 + \lambda\|\boldsymbol{z}'\|_F^2 = (\boldsymbol{\Phi}^{\top}\boldsymbol{\Phi} + \lambda\mathbf{I})^{-1}\boldsymbol{\Phi}^{\top}\boldsymbol{\xi}, \quad (9)$$

where $\boldsymbol{\Phi} \in \mathbb{R}^{T \times T_c}$ is the uniform B-spline basis matrix (a pre-defined hyperparameter determined by the knot placement and spline degree), acting as a linear temporal decoder that enforces smoothness constraint via the spline degree settings. The inferred control points $\boldsymbol{z}$ serve as a compact, fixed-length parametrization for quantization, converting variable-horizon trajectories into a unified representation of length $T_c$. The complete solving process is in Sec. A.5.

#### 3.3.2. LOCAL METRIC DISCRETIZATION

To address the issue of structural mismatch, the second objective aims to discretize the continuous vectors $\boldsymbol{z}$ into specific topological neighborhoods defined by "action schemas", while rectifying the action manifold's local anisotropy.

**Coarse Schema Selection.** We first partition the continuous linear space into local neighborhoods by quantizing $\boldsymbol{z}$ with the Layer 1 codebook $\mathcal{C}^{(1)} = \{\boldsymbol{c}_k^{(1)}\}_{k=1}^{K}$. Each codeword is intended to act as a topological anchor, defining the center of a local Voronoi region. To enable differentiable switching between these local charts, we compute the soft-assignment probability $p_k(\boldsymbol{z})$ using a temperature-scaled softmax:

$$p_k(\boldsymbol{z}) = \frac{\exp(-\|\boldsymbol{z} - \boldsymbol{c}_k^{(1)}\|^2/\tau)}{\sum_{j=1}^{K} \exp(-\|\boldsymbol{z} - \boldsymbol{c}_j^{(1)}\|^2/\tau)}. \quad (10)$$

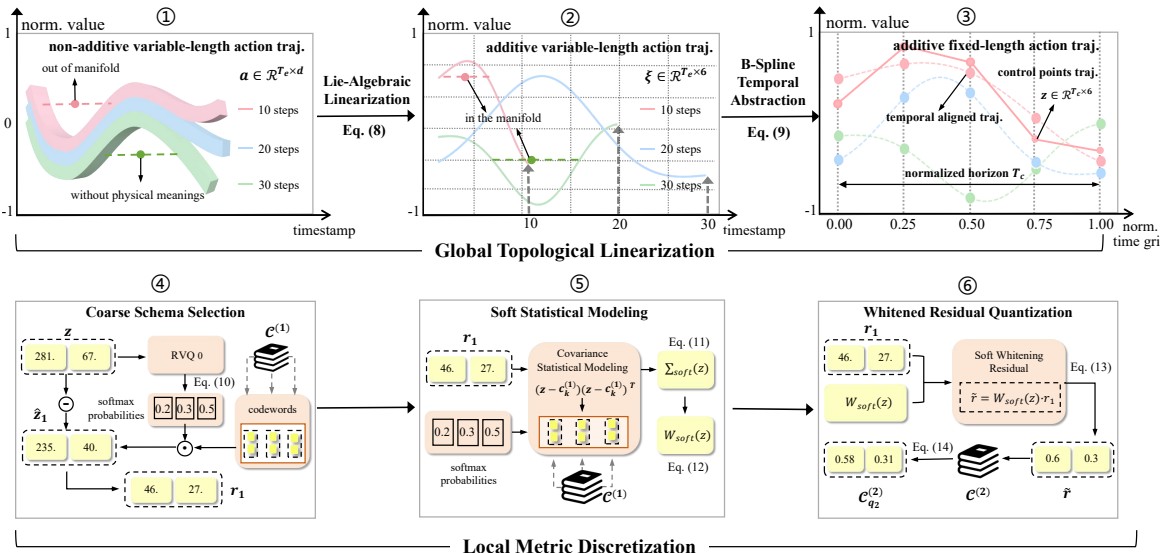

*Figure 2.* **Overview of LAST Tokenization Pipeline.** In Global Topological Linearization, the variable-length action trajectories are first linearized into the Lie-algebraic tangent space and then abstracted into fixed-length B-spline control points. In Local Metric Discretization, the control points are first coarsely quantized to select a global schema, followed by a covariance-aware whitening process that rectifies anisotropic residuals for optimal fine-grained quantization.

The coarse reconstruction $\hat{z}^{(1)} = \sum_{k=1}^{K} p_k(z) c_k^{(1)}$ represents the barycenter of the selected local neighborhood.

**Soft Statistical Modeling.** Within each local chart defined by an anchor $k$, the residual errors $r = z - \hat{z}^{(1)}$ exhibit highly anisotropic distributions specific to that neighborhood. We maintain a running estimate of this local covariance $\Sigma_k$ during training. Instead of a hard lookup, we compute a Soft Whitening Matrix $\mathcal{W}_{soft}$ by interpolating these local statistics based on the soft assignments. This ensures the metric transformation is smooth across the boundaries of local charts:

$$\Sigma_{soft}(z) = \sum_{k=1}^{K} p_k(z) \left[ (z - c_k^{(1)})(z - c_k^{(1)})^\top + \epsilon I \right],$$
(11)
$$\mathcal{W}_{soft}(z) = \Sigma_{soft}(z)^{-1/2}.$$
(12)

Here, $\mathcal{W}_{soft}$ acts as a local metric tensor correction, normalizing the curvature within the specific neighborhood.

**Whitened Residual Quantization.** We project the raw residual into the rectified local frame via the affine whitening transformation:

$$\tilde{r} = \mathcal{W}_{soft}(z) \cdot (z - \hat{z}^{(1)}).$$
(13)

In this rectified frame, the residual $\tilde{r}$ is approximately isotropic, i.e., $\tilde{r} \sim \mathcal{N}(0, I)$. We therefore *define* the Layer-2 quantizer to operate in this whitened coordinate system, where Euclidean distance is a faithful proxy of likelihood under the rectified metric. Concretely, the Layer-2 codebook $\mathcal{C}^{(2)} = \{c_j^{(2)}\}$ is learned in the whitened space and is responsible for capturing the remaining fine-grained, isotropic

variations. As a result, we can perform standard vector quantization on $\tilde{r}$:

$$q_2 = \arg\min_j \|\tilde{r} - c_j^{(2)}\|_2^2.$$
(14)

The final high-fidelity representation is reconstructed by mapping back from the local frame to the global space:

$$\hat{z} = \hat{z}^{(1)} + \Sigma_{soft}(z)^{1/2} \cdot c_{q_2}^{(2)}.$$
(15)

This mechanism ensures that the quantization lattice adapts to the principal axes of the local physical variance. The final result provides supervision for VLA that preserves both spatial and temporal consistency.

### 3.3.3. UNIFIED VLA LEARNING PARADIGM

The training of the full system proceeds in two stages, designed to optimize the topological and metric alignments respectively.

**Stage 1: Tokenizer Pre-training.** We first train the tokenizer $\mathcal{T}$ to minimize a compound objective consisting of reconstruction loss and commitment loss. The reconstruction loss $\mathcal{L}_{rec}$ measures fidelity in the physical tangent space, ensuring the geometric validity of the composed trajectory:

$$\mathcal{L}_{rec} = \mathbb{E}_{z \sim \mathcal{D}} \left[ \|z - \hat{z}\|_F^2 \right].$$
(16)

The commitment loss $\mathcal{L}_{com}$ operates in the whitened latent space. It constrains the whitened residuals to cluster around the codebook vectors, ensuring the codebook matches the isotropic distribution:

$$\mathcal{L}_{com} = \left( \| \operatorname{sg}[\tilde{r}] - c_{q_2}\|_2^2 + \beta \|\tilde{r} - \operatorname{sg}[c_{q_2}]\|_2^2 \right),$$
(17)

where sg[·] is the stop-gradient operation. The tokenizer objective is $\mathcal{L}_{tok} = \mathcal{L}_{rec} + \lambda_1 \mathcal{L}_{com}$, where $\lambda_1$ denotes the weighting coefficient.

**Stage 2: VLA Dual-Objective Training.** The VLA policy $\pi_\theta$ bridges vision-language inputs $(o, l)$ to actions via a dual-head architecture. Let $h$ be the final-layer hidden state at the action step shared by both heads: an AR head predicts discrete action tokens to capture discretized coarse structure, while a diffusion head generates continuous actions to refine fine-grained control.

(1) *Discrete Head (Topological Alignment):* The supervision targets $\bar{q}$ are obtained by encoding the ground-truth trajectory $a$ using the frozen **LAST** tokenizer $\mathcal{T}$. Specifically, the coarse schema token is obtained by selecting the mode of the soft distribution: $\bar{q}^{(1)} = \arg\max_k p_k(z)$. The subsequent residual tokens $\bar{q}^{(2)}$ are obtained via whitened residual quantization. Optimizing $\mathcal{L}_{dis}$ forces the model to select the correct topological schema, aligning the semantic intent with the global action topology:

$$\mathcal{L}_{dis} = -\mathbb{E}_{(s,\bar{q})\sim\mathcal{D}} \left[ \sum_l \log P_\theta(\bar{q}^{(l)} \mid s) \right]. \quad (18)$$

(2) *Continuous Head (Metric Refinement):* The default **LAST** model attaches a Diffusion Transformer (DiT) head to hidden states $h$, predicting the continuous action chunk via iterative denoising on whitened Lie-algebraic actions. The training objective is the standard noise-prediction MSE:

$$\mathcal{L}_{con} = \mathbb{E}_{a_0, \epsilon\sim\mathcal{N}(0,\mathbf{I}), k} \left[ \|\epsilon - \epsilon_\theta(a_k, k, h)\|_2^2 \right], \quad (19)$$

where $a_k$ is the noisy action at diffusion step $k$. For lightweight ablations, we also evaluate an MLP regression head that directly regresses $\phi(h)$ to $a$ via mean squared error (see Tab. 5a); the default **LAST**-Continuous model uses the DiT denoising head.

The total objective $\mathcal{L}_{vla} = \mathcal{L}_{dis} + \lambda_2 \mathcal{L}_{con}$ unifies intent selection with dynamic execution, where $\lambda_2$ denotes the weighting coefficient.

# 4. Experiments

## 4.1. Implementation Details

In real-world experiments, the backbone $\pi_\theta$ is a purely autoregressive Transformer with **3B** parameters, following the **Qwen2.5** configuration (Bai et al., 2025). The original action horizons in LIBERO, SimplerEnv, and real-world experiments are set to 8, 16, and 30 frames, respectively. For tokenizer pretraining, we train for 5 epochs to update the codebooks with loss weight $\lambda_1$=0.5, using a B-spline temporal basis with $T_c$=16 and a codebook size of $K$=1024. For VLA fine-tuning, we train for 5 epochs with a batch size

*Table 1.* **Reconstruction and Compression Comparison.**

| Method | Space | MAE ($\downarrow$) | CR ($\uparrow$) | CU ($\uparrow$) |
|---|---|---|---|---|
| Binning | Euclidean | 4.2e-2 | 1.0$\times$ | N/A |
| FAST | Frequency | 1.8e-2 | 4.0$\times$ | 78% |
| RVQVAE | Euclidean | 1.1e-2 | 7.4$\times$ | 51% |
| **LAST (Ours)** | **Tangent** | **0.6**e-**2** | **8.0**$\times$ | **96**% |

of 256 on 8$\times$A800 GPUs, using loss weight $\lambda_2$=10. Additional hyperparameter selections are provided in Sec. A.4.

## 4.2. Benchmarks

Comprehensive descriptions of all task definitions, data collection protocols, and environment settings are provided in Sec. A.2. **Real-World Benchmarks.** We evaluate three real-world tasks that emphasize different aspects of manipulation: PlaceObj, ZipSeal, and TubeRack. Concretely, we collect 200 demonstrations per task on AgileX Cobot Magic platform, sampled at 30 Hz. Each trajectory spans 400 $\sim$ 600 frames and logs the full end-effector space control signals for both arms. Synchronized RGB-D imagery is recorded from three viewpoints: a first-person, top-down egocentric view using an Intel RealSense D455, and two wrist-mounted views using Intel RealSense D435 cameras.

**Simulation Benchmarks.** *LIBERO* (Liu et al., 2024) comprises four task suites: spatial, object, goal, and long-horizon compositional. Each suite contains 10 robotic manipulation tasks, with 50 demonstrations provided for each task. *SimplerEnv* (Li et al., 2024a) reflects the performance of real-world policies by replicating physical dynamics and visual appearance, encompassing diverse variations in lighting, textures, and viewpoints.

## 4.3. Advantages of Manifold Alignment

We evaluate the reconstruction quality and compression efficiency of **LAST** against representative tokenization baselines: (i) **Dimension-wise Binning** (Kim et al., 2024); (ii) **Frequency-domain BPE** (Pertsch et al., 2025); and (iii) **Standard RVQ-VAE** (Lee et al., 2024). We report three key metrics: (1) **Mean Absolute Error (MAE)**; (2) **Compression Ratio (CR)**; and (3) **Codebook Utilization (CU)**. The detailed definitions are in Sec. A.6.

As summarized in Tab. 1, **LAST** achieves a Pareto-optimal balance between precision and compression. By operating in the Lie-algebraic tangent space, **LAST** avoids the geometric distortions inherent in Euclidean baselines. This allows it to achieve a reconstruction error (MAE) of **0.6**e-**2**, which is nearly **50**% lower than the Euclidean RVQVAE (1.1e-2) and **3**$\times$ better than FAST (1.8e-2), proving that manifold alignment is crucial for high-fidelity representation. Meanwhile, the proposed *Covariance-Aware Whitening* mechanism significantly boosts Codebook Utilization (**96**% vs. 51%). Standard RVQVAE suffers from severe

*Table 2.* **Experimental Results for the LIBERO Benchmarks.** Best results are bolded.

| Model | Spatial | | Object | | Goal | | Long | | Average | |
|---|---|---|---|---|---|---|---|---|---|---|
| | SR (↑) | Rank (↓) | SR (↑) | Rank (↓) | SR (↑) | Rank (↓) | SR (↑) | Rank (↓) | SR (↑) | Rank (↓) |
| Octo (Octo Model Team et al., 2023) | 78.9% | 11 | 85.7% | 11 | 84.6% | 9 | 51.1% | 11 | 75.1% | 11 |
| OpenVLA (Kim et al., 2024) | 84.7% | 10 | 88.4% | 10 | 79.2% | 10 | 53.7% | 10 | 76.5% | 10 |
| SpatialVLA (Qu et al., 2025) | 88.2% | 9 | 89.9% | 9 | 78.6% | 11 | 55.5% | 9 | 78.1% | 9 |
| GR00T-N1.5 (NVIDIA et al., 2025) | 92.0% | 8 | 92.0% | 8 | 86.0% | 8 | 76.0% | 7 | 86.5% | 7 |
| $\pi_0$ (Black et al., 2024) | 96.8% | 2 | **98.8%** | **1** | 95.8% | 2 | 85.2% | 6 | 94.1% | 2 |
| $\pi_0$-FAST (Pertsch et al., 2025) | 96.4% | 4 | 96.8% | 6 | 88.6% | 7 | 60.2% | 8 | 85.5% | 8 |
| $\pi_0$-FAST-Continuous | 96.6% | 3 | 97.2% | 4 | 93.1% | 5 | 89.4% | 2 | 94.1% | 2 |
| BEAST (Zhou et al., 2025) | 92.9% | 7 | 97.5% | 3 | 93.1% | 5 | 86.4% | 3 | 92.5% | 6 |
| BEAST-Continuous | 94.1% | 5 | 96.8% | 6 | 95.1% | 3 | 85.7% | 5 | 92.9% | 5 |
| **LAST-Discretized** | 94.1% | 5 | 98.7% | 2 | 94.6% | 4 | 86.0% | 4 | 93.4% | 4 |
| **LAST-Continuous** | **98.4%** | **1** | 97.0% | 5 | **96.7%** | **1** | **91.2%** | **1** | **95.8%** | **1** |

*Table 3.* **Evaluation on SimplerEnv−WidowX across diverse manipulation tasks.**

| Model | Put Spoon | | Put Carrot | | Stack Block | | Put Eggplant | | Overall |
|---|---|---|---|---|---|---|---|---|---|
| | Grasp | Success | Grasp | Success | Grasp | Success | Grasp | Success | Success |
| Octo-Base (Octo Model Team et al., 2023) | 34.7% | 12.5% | 52.8% | 8.3% | 31.9% | 0.0% | 66.7% | 43.1% | 16.0% |
| $\pi_0$ (Black et al., 2024) | - | 29.1% | - | 0.0% | - | 16.6% | - | 62.5% | 27.1% |
| Octo-Small (Octo Model Team et al., 2023) | 77.8% | 47.2% | 27.8% | 9.7% | 40.3% | 4.2% | 87.5% | 56.9% | 29.5% |
| RoboVLMs (Li et al., 2024b) | 70.8% | 45.8% | 33.3% | 20.8% | 54.2% | 4.2% | 91.7% | 79.2% | 37.5% |
| OpenVLA-OFT (Kim et al., 2025) | - | 34.5% | - | 30.0% | - | 30.0% | - | 72.5% | 41.8% |
| SpatialVLA (Qu et al., 2025) | 20.8% | 16.7% | 29.2% | 25.0% | 62.5% | 29.2% | 100% | 100% | 42.7% |
| $\pi_0$-FAST (Pertsch et al., 2025) | 33.3% | 29.2% | 25.0% | 20.8% | 37.5% | 12.5% | 75.0% | 66.7% | 32.3% |
| $\pi_0$-FAST-Continuous | 70.8% | 45.8% | 66.7% | **41.7%** | 58.3% | 25.0% | 79.2% | 70.8% | 45.8% |
| BEAST (Zhou et al., 2025) | 66.7% | 41.7% | 37.5% | 25.0% | 50.0% | 20.8% | 87.5% | 75.0% | 37.5% |
| BEAST-Continuous | 54.2% | 37.5% | 70.8% | **41.7%** | 33.3% | 12.5% | 87.5% | 70.8% | 40.6% |
| **LAST-Discretized** | 83.3% | 58.3% | **79.2%** | 37.5% | **83.3%** | 29.2% | **100.0%** | **95.8%** | 55.2% |
| **LAST-Continuous** | **87.5%** | **62.5%** | **79.2%** | **41.7%** | 79.2% | **33.3%** | **100.0%** | 91.7% | **57.3%** |

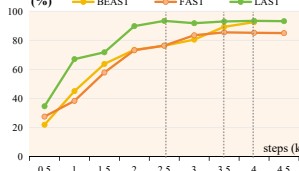

*Figure 3.* **Training Convergence on LIBERO.**

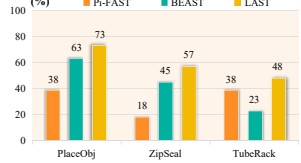

*Figure 4.* **Real-World Evaluation Comparison.**

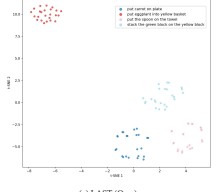
(a) LAST (Ours).

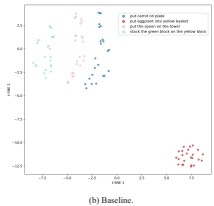
(b) Baseline.

*Figure 5.* **t-SNE Visualization Comparison.**

"dead codes" because it attempts to fit spherical clusters to highly anisotropic robot data. In contrast, **LAST** rectifies the latent space into an isotropic distribution, enabling more codewords to contribute to describing the motion manifold.

To further illustrate the impact of these tokenizer-level gains, we compare end-to-end learning dynamics on the LIBERO benchmark. We report the average performance results in Fig. 3 and sub-task results in Fig. 9. Results indicate that **LAST** consistently attains higher success rates and reaches its performance plateau earlier at 2.5k steps (compared to 3.5k for $\pi_0$-FAST and 4k for BEAST). This accelerated convergence can be attributed to the discrete action intent induced by **LAST**. It simplifies the optimization landscape, allowing the downstream policy to learn complex manipulation tasks more efficiently.

### 4.4. Gains in Generalist VLA Learning

In tokenizer-based methods, "Discretized" refers to using predicted tokens directly for action generation, while "Continuous" refers to generating actions based on trained hidden states with an additional Diffusion Transformer (DiT) head,

allowing for finer control over action generation.

**Simulation Benchmarks Evaluation.** On **LIBERO** (Table 2), **LAST**-Continuous achieves a new state-of-the-art average success rate of **95.8%**, outperforming both $\pi_0$-FAST-Continuous and $\pi_0$ (both at 94.1%). Notably, while continuous refinement generally improves performance across all tokenizers (e.g., FAST improves from 85.5% to 94.1%), **LAST** derives the most significant benefit in long-horizon tasks (improving from 86.0% to 91.2%), confirming that our metric-aligned latent space provides a superior foundation for fine-grained trajectory regression. Even in its discretized form, **LAST** remains highly competitive, ranking fourth overall. On **SimplerEnv** (Table 3), **LAST** demonstrates exceptional generalization. **LAST**-Continuous achieves the highest overall success rate of **57.3%**, surpassing both $\pi_0$-FAST-Continuous (45.8%) and OpenVLA-OFT (41.8%). It achieves strong performance on the Put Eggplant task (91.7% success) while maintaining competitive results on precise pick-and-place tasks like Put Spoon (62.5%). This validates that the structural isomorphism established by **LAST** facilitates robust sim-to-real transfer and resilience to visual perturbations.

*Table 4.* **Ablations of LAST Tokenizer Components.**

*(a)* **Effects of Manifold Alignment.**

| $\mathfrak{se}(3)$ | $\mathcal{W}$ | Spatial | Object | Goal | Long |
|---|---|---|---|---|---|
| ✗ | ✓ | 88.5 | 96.1 | 93.8 | 83.4 |
| ✓ | ✗ | 92.8 | 93.5 | 89.2 | 84.1 |
| ✓ | ✓ | **94.1** | **98.7** | **94.6** | **86.0** |

*(b)* **Effects of Tokenizer Training Objectives.**

| Setting | Spatial | Object | Goal | Long |
|---|---|---|---|---|
| w/o $\mathcal{L}_{com}$ ($\lambda_1=0$) | 82.4 | 93.6 | 89.1 | 66.2 |
| dominant $\mathcal{L}_{com}$ ($\lambda_1=10$) | 51.8 | 38.7 | 73.5 | 26.4 |
| Full ($\lambda_1=0.5$) | **94.1** | **98.7** | **94.6** | **86.0** |

*Table 5.* **Ablations of VLA Policy Architecture and Objectives.**

*(a)* **Effect of Continuous Head Architecture.**

| Continuous Head | PlaceObj | ZipSeal | TubeRack |
|---|---|---|---|
| MLP-Net | **80%** | 52% | 43% |
| DiT-Block | 73% | **57%** | **48%** |

*(b)* **Effect of Discrete Supervision.**

| Discrete Sup. ($\mathcal{L}_{dis}$) | PlaceObj | ZipSeal | TubeRack |
|---|---|---|---|
| ✗ (w/o $\mathcal{L}_{dis}$) | 45% | 22% | 18% |
| ✓ (w/ $\mathcal{L}_{dis}$) | **73%** | **57%** | **48%** |

**Qualitative Analysis.** Figure 5 presents the t-SNE visualization comparison of the hidden states in the four tasks of the SimplerEnv dataset. Panel (a) shows the results of **LAST**, where distinct action clusters are clearly formed. In contrast, Panel (b) demonstrates the Baseline (without **LAST** supervision), where the tasks exhibit more mixed clusters. This visual difference suggests that **LAST** learns more coherent and task-specific representations, providing stronger separability in the latent space. Such good separability lays the foundation for effective GW alignment.

**Real-World Evaluation.** As illustrated in Figure 4, **LAST** achieves the best performance among the compared methods on our real-world benchmark, with success rates of 73%, 57%, and 48% on PlaceObj, ZipSeal, and TubeRack, respectively, surpassing both $\pi_0$-FAST and BEAST. The performance gap highlights specific theoretical deficiencies in prior methods: $\pi_0$-FAST suffers catastrophic failure on the contact-rich ZipSeal task (18%), confirming that global frequency-domain compression introduces detrimental Gibbs ringing artifacts during abrupt motion changes, while BEAST struggles with the high-precision TubeRack task (23%), validating that Euclidean interpolation on the curved $SE(3)$ manifold causes metric distortions. In contrast, by leveraging Lie-algebraic manifold alignment for geometric validity and covariance-aware whitening for statistical efficiency, **LAST** effectively mitigates these issues, enabling robust control in both deformable manipulation and precise insertion scenarios. The real-world comparison in Fig. 4 uses the DiT-Block as the default continuous head.

### 4.5. Ablation of Theoretical Components

We verify the critical design choices in both the **LAST** tokenizer and the downstream VLA policy.

**Effects of Manifold Alignment.** Tab. 4a confirms that our theoretical pillars are essential. Mapping to $\mathfrak{se}(3)$ ensures geometric validity, crucial for Spatial tasks (+5.6%). Covariance-Aware Whitening ($\mathcal{W}$) prevents codebook col-

lapse in anisotropic dimensions, boosting precision in Object task (+5.2%) and Goal task (+5.4%).

**Effects of Tokenizer Objectives.** Tab. 4b highlights the balance between fidelity and structure. Removing the commitment loss ($\lambda_1=0$) causes the discrete codes to fail to cluster into stable schemas, so the VLA loses its "topological anchor" and degrades noticeably on **Long** horizon tasks (66.2%). Conversely, dominating the loss with the commitment term ($\lambda_1=10$) suppresses the reconstruction gradient: the discrete tokens lose their physical grounding and performance plummets (*e.g.,* 26.4% on Long).

**Effects of Continuous Head Architecture.** In Tab. 5a, we compare using a simple MLP versus a DiT block for the continuous refinement head. Interestingly, the **MLP** baseline matches or slightly exceeds DiT on the simpler PlaceObj task (80% vs. 73%), supporting our theory that the schema prior collapses the search space into a simple local convex basin, making it easy to learn even for simple networks. However, the **DiT** head provides clear gains on contact-rich and high-precision tasks (ZipSeal +5%, TubeRack +5%), as it better models the residual uncertainty and high-frequency contact dynamics.

**Necessity of Discrete Schema Supervision.** Tab. 5b provides the strongest evidence for our *"topological collapse"* hypothesis. When we remove the discrete supervision ($\mathcal{L}_{dis}$) and train the model via direct regression (even with a DiT head), performance plummets (*e.g.,* ZipSeal drops from 57% to 22% and TubeRack drops from 48% to 18%). This confirms that without explicitly predicting the action schema to lock the target basin, the continuous head fails to achieve high-precision manipulation.

**Hyperparameter Analysis.** We perform sensitivity analysis on key hyperparameters, including $\lambda_1$ for commitment regularization (Tab. 10), $\lambda_2$ for planning-execution balance (Tab. 11), $K$ for codebook size (Tab. 9), and $\beta$ for latent commitment (Tab. 8), to assess their impact on performance.

# 5. Conclusion

We introduced **LAST** as a mathematically rigorous interface for the Action Modality. By leveraging Global Topological Linearization and Local Metric Discretization, **LAST** resolves geometric and statistical mismatches in prior tokenizers. This results in a physically consistent latent space that transforms non-convex global search into local convex refinement, establishing the foundation for Gromov-Wasserstein Alignment between semantic and action space.

# Impact Statement

This paper advances generalist embodied intelligence via geometry-aware alignment between vision-language and action manifolds. We highlight two analytical considerations that bound the applicability of our framework. *(i) Whitening complexity.* Local Metric Discretization estimates per-schema covariances at $\mathcal{O}(Kd^3)$ cost ($K$: codebook size, $d$: residual dimensionality), negligible at our scale ($K=1024$, $d=6$); high-DoF embodiments such as dexterous multi-finger control may require low-rank or shrinkage estimators to avoid instability from under-sampled covariance modes. *(ii) Local approximation validity.* Global Topological Linearization relies on the $SE(3)$ logarithm map, whose injectivity degrades near the cut locus (rotations approaching $\pi$). At $\geq 30$ Hz sampling, inter-frame residuals stay within the injectivity radius and linearization is essentially distortion-free; sparsely sampled or aggressively rotating trajectories may need multi-chart covers or recursive linearization.

# Acknowledgements

This work was supported by the National Natural Science Foundation of China under Grants U23A20387, 62502518, 62532003, U25A20536, in part by the Pengcheng Laboratory Research Project under Grant PCL2023A08, and also in part by the Postdoctoral Fellowship Program of CPSF under Grant Number GZC20251036.

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

# A. Appendix

This supplementary material provides detailed theoretical proofs, experimental settings, and additional analysis to support the claims made in the main manuscript. The appendix is organized as follows:

- **Sec. A.1**: Detailed mathematical proofs for the lemmas regarding Statistical Mismatch, Mode Averaging, and Topological Collapse.

- **Sec. A.2**: Detailed descriptions of real-world and simulation benchmarks.

- **Sec. A.3**: Implementation details of the VLA architecture, specifically the Dual-Head mechanism, and training hyperparameters.

- **Sec. A.4**: Supplementary ablation studies on tokenizer hyperparameters (*e.g.,* chunk length, codebook size, loss weights).

- **Sec. A.5**: Algorithmic details of the B-spline parameterization.

- **Sec. A.6**: Definitions of the evaluation metrics.

## A.1. Theoretical Proofs and Derivations

In this section, we provide rigorous proofs for the theoretical foundations of the **LAST** framework. Our analysis follows a logical progression: we first prove the inherent failure of Euclidean regression on non-convex action manifolds (Lemma A.1), then demonstrate the optimization instability caused by statistical anisotropy (Lemma A.2), and finally justify the efficiency of our hybrid architecture via complexity analysis (Lemma A.3). Lemma A.4 further establishes the intra-cluster additivity of our Lie-algebraic B-spline representation used in the global linearization stage.

**Lemma A.1** (Mode Averaging on Non-Convex Manifolds). *For a multi-modal conditional action distribution $P(\boldsymbol{a}|\boldsymbol{s})$, the Mean Squared Error (MSE) estimator $\Phi^*$ converges to a convex combination of modes that lies in the convex hull of the manifold $\mathrm{Conv}(\mathcal{M})$ but generally falls into invalid regions outside $\mathcal{M}$ itself.*

**Proof.** Consider a multi-modal action distribution modeled as a Gaussian Mixture: $P(\boldsymbol{a}|\boldsymbol{s}) = \sum_{k=1}^{K} \omega_k(\boldsymbol{s})\mathcal{N}(\boldsymbol{a}; \boldsymbol{\mu}_k, \boldsymbol{\Sigma}_k)$, where $\boldsymbol{\mu}_k \in \mathcal{M}$ are distinct valid action centroids. The standard regression objective minimizes $\mathcal{L}(\Phi) = \mathbb{E}_{\boldsymbol{s}}\mathbb{E}_{\boldsymbol{a}\sim P(\boldsymbol{a}|\boldsymbol{s})}[\|\Phi(\boldsymbol{s}) - \boldsymbol{a}\|^2]$. The optimal estimator is the conditional expectation:

$$\Phi^*(\boldsymbol{s}) = \int \boldsymbol{a}P(\boldsymbol{a}|\boldsymbol{s})d\boldsymbol{a} = \sum_{k=1}^{K} \omega_k(\boldsymbol{s})\,\boldsymbol{\mu}_k. \tag{20}$$

By definition, $\Phi^*(\boldsymbol{s})$ is a convex combination of valid points and thus $\Phi^*(\boldsymbol{s}) \in \mathrm{Conv}(\mathcal{M})$.

However, robotic action manifolds are strictly non-convex due to kinematic limits and obstacle-induced "barriers" $\mathcal{B} = \mathbb{R}^N \setminus \mathcal{M}$. For any non-convex set, $\mathrm{Conv}(\mathcal{M}) \setminus \mathcal{M} \neq \emptyset$. Specifically, if two modes $\boldsymbol{\mu}_i, \boldsymbol{\mu}_j$ represent trajectories separated by an obstacle with distance $\delta$, the projection of the estimator onto the valid manifold satisfies:

$$\min_{\boldsymbol{a}\in\mathcal{M}} \|\Phi^*(\boldsymbol{s}) - \boldsymbol{a}\| \geq \min_k(1 - \omega_k(\boldsymbol{s}))\frac{\delta}{2} > 0. \tag{21}$$

This implies $\Phi^*(\boldsymbol{s}) \in \mathcal{B}$, meaning the regression result collapses into high-energy forbidden zones (*e.g.,* collisions) rather than staying on the manifold surface. **LAST** resolves this by using discrete tokens to select a single mode before performing continuous refinement. $\square$

**Lemma A.2** (Statistical Mismatch and Lipschitz Sensitivity). *Let the semantic space $\mathcal{X}$ be statistically isotropic ($G_{\mathcal{X}} \approx I$) and the action manifold $\mathcal{Y}$ be highly anisotropic ($G_{\mathcal{Y}} \neq I$). A direct mapping $\Phi : \mathcal{X} \rightarrow \mathcal{Y}$ requires a Lipschitz constant $\mathrm{Lip}(\Phi)$ proportional to the square root of the condition number $\kappa(G_{\mathcal{Y}})$, leading to gradient instability.*

**Proof.** Modern vision-language backbones typically normalize features onto a hypersphere via Layer Normalization, rendering the semantic space $\mathcal{X}$ statistically isotropic with covariance $\Sigma_{\mathcal{X}} \approx I$. In contrast, the action manifold $\mathcal{Y}$ is

governed by physical constraints, resulting in a highly anisotropic covariance $\Sigma_{\mathcal{Y}}$ where variances along task-relevant axes ($\lambda_{\min}$) are significantly smaller than along task-redundant axes ($\lambda_{\max}$).

Let $\boldsymbol{u}, \boldsymbol{v} \in \mathcal{X}$ be two semantic vectors separated by $\delta = \|\boldsymbol{u} - \boldsymbol{v}\|_2$. For a mapping $\Phi$ to transform the isotropic distribution in $\mathcal{X}$ to the anisotropic target in $\mathcal{Y}$ (equipped with metric tensor $G_{\mathcal{Y}} = \Sigma_{\mathcal{Y}}^{-1}$), the Jacobian $\mathbf{J}_\Phi = \frac{\partial \Phi}{\partial \boldsymbol{u}}$ must locally satisfy $\mathbf{J}_\Phi \mathbf{J}_\Phi^\top \approx \Sigma_{\mathcal{Y}}$ (Arvanitidis et al., 2017). Using the eigendecomposition $\Sigma_{\mathcal{Y}} = U\Lambda U^\top$, the Lipschitz constant $\text{Lip}(\Phi)$ is bounded by the maximum stretching required to cover the manifold's high-variance dimensions relative to its constraints:

$$\text{Lip}(\Phi) = \sup \frac{d_{\mathcal{Y}}(\Phi(\boldsymbol{u}), \Phi(\boldsymbol{v}))}{\|\boldsymbol{u} - \boldsymbol{v}\|_2} \propto \sqrt{\frac{\lambda_{\max}}{\lambda_{\min}}} = \sqrt{\kappa(\Sigma_{\mathcal{Y}})} = \sqrt{\kappa(G_{\mathcal{Y}})}. \tag{22}$$

Optimization stability is governed by the condition number of the loss Hessian, $\kappa(\nabla^2 \mathcal{L}) \approx \kappa(\mathbf{J}_\Phi^\top \mathbf{J}_\Phi) = \kappa(\Sigma_{\mathcal{Y}})$. Since robotic action manifolds often exhibit $\kappa(\Sigma_{\mathcal{Y}}) \gg 10^3$, $\Phi$ must be highly non-smooth. This ill-conditioned landscape leads to gradient explosion or vanishing, necessitating structural alignment (*e.g.,* whitening) provided by **LAST**. $\qquad\square$

**Lemma A.3** (Topological Complexity Collapse). *Partitioning the action manifold $\mathcal{M}$ into $K$ local charts indexed by discrete intents reduces the search complexity (metric entropy) from a linear dependence on the ambient dimension $N$ to a linear dependence on the intrinsic dimension $d$, where $d \ll N$.*

**Proof.** We measure complexity using the *log-covering number* $\log \mathcal{N}(\epsilon, \Omega)$, which scales with the number of $\epsilon$-balls needed to cover $\Omega$.

Without a hybrid architecture, a model approximates the manifold $\mathcal{M}$ within the high-dimensional ambient space $\mathbb{R}^N$ (where $N = T \cdot d$, i.e., horizon times per-step action dimensionality). The metric entropy scales as:

$$\log \mathcal{N}_{global} \approx N \log(R/\epsilon), \tag{23}$$

where $R$ is the manifold diameter. This reflects the "curse of dimensionality" where search complexity grows linearly with the observation dimension $N$.

**LAST** partitions $\mathcal{M}$ into $K$ local charts $\{\mathcal{U}_k\}_{k=1}^K$. Each chart is locally diffeomorphic to a low-dimensional Euclidean space $\mathbb{R}^d$, representing the intrinsic motion parameters. The total complexity of the **LAST** framework is:

$$\mathcal{C}_{\textbf{LAST}} = \underbrace{\log K}_{\text{Intent Selection}} + \max_k \underbrace{\log \mathcal{N}(\epsilon, \mathcal{U}_k)}_{\text{Refinement}}. \tag{24}$$

Since refinement occurs in the intrinsic space (*e.g.,* Lie-algebraic tangent space):

$$\log \mathcal{N}(\epsilon, \mathcal{U}_k) \approx d \log(r/\epsilon), \tag{25}$$

where $r \ll R$ is the local cluster radius. The total complexity $\mathcal{C}_{\textbf{LAST}} \approx \log K + d \log(r/\epsilon)$ signifies a **collapse** from $N$-dependence to $d$-dependence. This justifies the efficiency of anchoring the diffusion head to discrete predicted tokens. $\quad\square$

**Lemma A.4** (Intra-Cluster Additivity of Lie-Algebraic B-Spline Representation). *Consider a cluster (local chart) on $SE(3)$ where all relative motions w.r.t. a reference pose stay within a small neighborhood such that the logarithmic map is well-defined. Let $\boldsymbol{\xi}_t = \log(P_1^{-1} P_t)^\vee \in \mathbb{R}^6$ be the Lie-algebraic coordinates in Eq. equation 8, and let the B-spline control-point representation be $\boldsymbol{z} = \mathcal{A}\boldsymbol{\xi}$ with $\mathcal{A} = (\boldsymbol{\Phi}^\top \boldsymbol{\Phi} + \lambda \mathbf{I})^{-1} \boldsymbol{\Phi}^\top$ in Eq. equation 9. Then, within the cluster, the representation is additive up to second-order Lie linearization error: (i) for small motions, group composition corresponds to tangent-space addition with $\mathcal{O}(\varepsilon^2)$ error; (ii) the B-spline abstraction is a linear map and thus preserves additivity exactly in the tangent space.*

**Proof.** The overall process consists of two steps:

Step 1: Tangent-space additivity via BCH. Fix a reference pose $P_1$ (or a cluster-specific reference $\bar{P}_k$) and define relative transforms $G = P_1^{-1} P \in SE(3)$ with coordinates $\boldsymbol{\xi} = \log(G)^\vee \in \mathbb{R}^6$. Consider two motions within the same cluster neighborhood:

$$G_a = \exp(\boldsymbol{\xi}_a^\wedge), \qquad G_b = \exp(\boldsymbol{\xi}_b^\wedge), \tag{26}$$

where $\|\boldsymbol{\xi}_a\|_2, \|\boldsymbol{\xi}_b\|_2 \leq \varepsilon$ and $\varepsilon$ is small enough to remain inside the injectivity radius of $\log(\cdot)$. By the Baker–Campbell–Hausdorff (BCH) formula,

$$\log(G_a G_b) = \log\big(\exp(\boldsymbol{\xi}_a^\wedge) \exp(\boldsymbol{\xi}_b^\wedge)\big) = (\boldsymbol{\xi}_a + \boldsymbol{\xi}_b)^\wedge + \frac{1}{2}[\boldsymbol{\xi}_a^\wedge, \boldsymbol{\xi}_b^\wedge] + \mathcal{O}(\varepsilon^3), \tag{27}$$

where $[\cdot, \cdot]$ is the Lie bracket. Applying $(\cdot)^\vee$ and using $\|[\boldsymbol{\xi}_a^\wedge, \boldsymbol{\xi}_b^\wedge]\| \leq C\|\boldsymbol{\xi}_a\|\,\|\boldsymbol{\xi}_b\|$ for some constant $C$ in a fixed chart, we obtain

$$\big\| \log(G_a G_b)^\vee - (\boldsymbol{\xi}_a + \boldsymbol{\xi}_b)\big\|_2 \leq C\|\boldsymbol{\xi}_a\|_2\|\boldsymbol{\xi}_b\|_2 + \mathcal{O}(\varepsilon^3) = \mathcal{O}(\varepsilon^2). \tag{28}$$

Thus, *within a cluster of small motions*, composing two motions in $SE(3)$ is equivalent to adding their tangent vectors up to second-order error.

This extends pointwise to trajectories. Let a trajectory be represented by relative transforms $\{G_t\}_{t=1}^T$ with $\boldsymbol{\xi}_t = \log(G_t)^\vee$. For two trajectories in the same cluster satisfying $\|\boldsymbol{\xi}_t^{(a)}\|_2, \|\boldsymbol{\xi}_t^{(b)}\|_2 \leq \varepsilon$ for all $t$, the pointwise composed motion $G_t^{(a)} G_t^{(b)}$ satisfies

$$\log\big(G_t^{(a)} G_t^{(b)}\big)^\vee = \boldsymbol{\xi}_t^{(a)} + \boldsymbol{\xi}_t^{(b)} + \mathcal{O}(\varepsilon^2), \qquad \forall t. \tag{29}$$

Hence the Lie-algebraic trajectory coordinates are *cluster-wise additive* up to $\mathcal{O}(\varepsilon^2)$.

Step 2: Exact additivity preserved by B-spline abstraction. Vectorize the tangent trajectory as $\boldsymbol{\xi} = [\boldsymbol{\xi}_1; \ldots; \boldsymbol{\xi}_T] \in \mathbb{R}^{6T}$. The ridge-regression solution in Eq. equation 9 defines a linear operator

$$\boldsymbol{z} = \mathcal{A}\boldsymbol{\xi}, \qquad \mathcal{A} = (\boldsymbol{\Phi}^\top \boldsymbol{\Phi} + \lambda \mathbf{I})^{-1} \boldsymbol{\Phi}^\top, \tag{30}$$

where $\boldsymbol{\Phi}$ and $\lambda$ are fixed. Therefore, for any $\boldsymbol{\xi}^{(a)}, \boldsymbol{\xi}^{(b)}$ and scalars $\alpha, \beta$,

$$\mathcal{A}(\alpha\boldsymbol{\xi}^{(a)} + \beta\boldsymbol{\xi}^{(b)}) = \alpha\mathcal{A}\boldsymbol{\xi}^{(a)} + \beta\mathcal{A}\boldsymbol{\xi}^{(b)}. \tag{31}$$

In particular, $\mathcal{A}(\boldsymbol{\xi}^{(a)} + \boldsymbol{\xi}^{(b)}) = \mathcal{A}\boldsymbol{\xi}^{(a)} + \mathcal{A}\boldsymbol{\xi}^{(b)}$, i.e., the B-spline control-point representation is *exactly additive* in the tangent space.

**Conclusion.** Combining Eq. equation 29 and Eq. equation 31, within each cluster the overall representation $a \mapsto \boldsymbol{\xi} \mapsto \boldsymbol{z}$ is additive up to the second-order Lie linearization error:

$$\boldsymbol{z}^{(a \circ b)} = \mathcal{A}\boldsymbol{\xi}^{(a \circ b)} = \mathcal{A}\big(\boldsymbol{\xi}^{(a)} + \boldsymbol{\xi}^{(b)}\big) + \mathcal{O}(\varepsilon^2) = \boldsymbol{z}^{(a)} + \boldsymbol{z}^{(b)} + \mathcal{O}(\varepsilon^2). \tag{32}$$

This proves the desired intra-cluster additivity of the proposed representation. $\qquad\square$

## A.2. Dataset and Benchmark Details

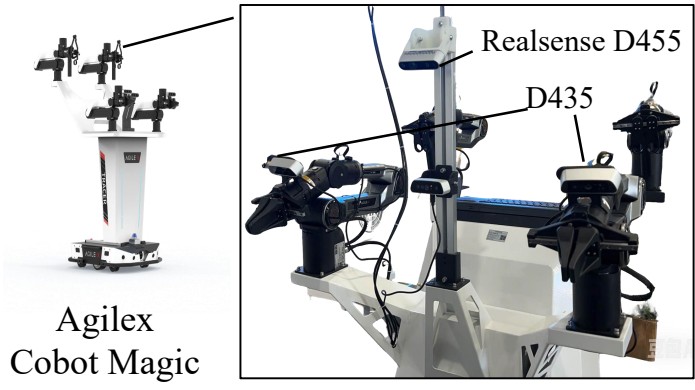

*Figure 6.* **Dual-arm data-collection platform.** We use an AgileX *Cobot Magic* base with two collaborative arms. RGB(-D) videos are captured from an overhead Intel RealSense D455 and wrist-mounted Intel RealSense D435i cameras; all streams are time-synchronized with joint-space commands at 30 Hz.

### A.2.1. REAL-WORLD BENCHMARKS

**Data Collection Platform.** We collect high-quality bimanual demonstrations using an AgileX *Cobot Magic* platform (Fig. 6). The hardware includes an overhead Intel RealSense D455 for global observation and two wrist-mounted D435i cameras for local interaction details. All data, including 7-DoF joint commands and multi-view video, are synchronized

*Table 6.* Task objectives and challenges in the real-world benchmark suite.

| Task | Objective | Key Challenges |
|------|-----------|----------------|
| PlaceObj | Place named object into basket. | Instruction grounding, Table-top grasping |
| ZipSeal | Align edges and close zipper. | Deformable control, Bimanual coordination |
| TubeRack | Insert tube into rack slot. | High spatial precision, Tight tolerances |

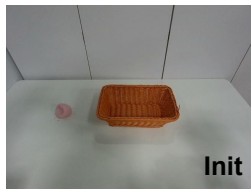 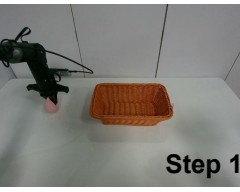 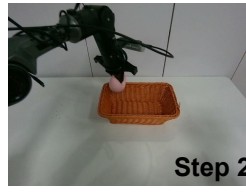 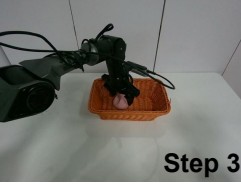 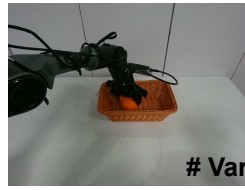

***PlaceObj:*** *"Place object on the table into the basket."* (Step1) Grasp the object on the table surface. (Step2) Lift it up above the basket. (Step3) Release the object. (# Var) Task varies on the object category.

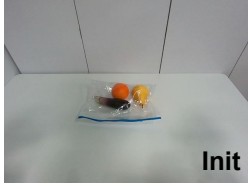 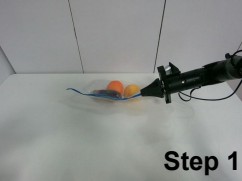 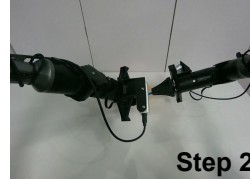 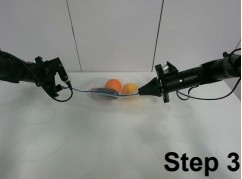 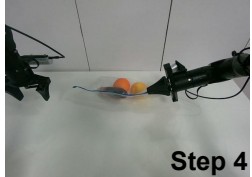

***ZipSeal:*** *"seal a plastic bag with zipper placed on the table."* (Step1) Grasp one side of the bag. (Step2) Grasp the zipper tab with the other gripper. (Step3) Pulling the zipper along the sealing track. (Step4) Release the plastic bag.

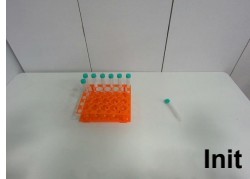 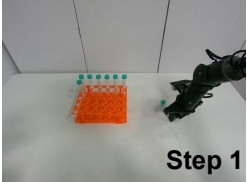 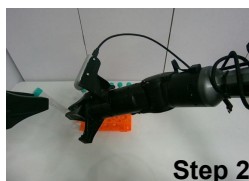 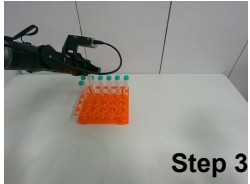 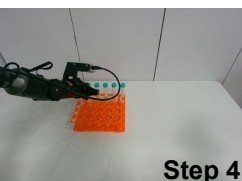

***TubeRack:*** *"Insert the tube on the table surface into an empty slot of the rack."* (Step1) Pick up the tube. (Step2) Reorient the tube and pass it to the other gripper. (Step3) Insert the tube into the rack. (Step4) Release the tube.

*Figure 7.* **Real-world tasks and step-wise rollouts. PlaceObj**: grasp, lift, and place. **ZipSeal**: bimanual alignment and closing along the track. **TubeRack**: pick, reorient, and insert with precision.

and recorded at 30 Hz. The final dataset comprises 200 demonstrations per task, totaling over 600 trajectories with lengths between 400 and 600 frames.

**PlaceObj: Semantic Grounding.** As illustrated in Fig. 7 (top), this task requires the robot to interpret natural language instructions to pick a specific object from distractors and place it into a basket. According to Table 6, the primary challenges involve instruction grounding under visual clutter and maintaining reliable tabletop grasping across varying object categories.

**ZipSeal: Dexterous Bimanual Control.** This task involves aligning the edges of a resealable bag and pulling a zipper along a predefined track (Fig. 7, middle). It serves as a benchmark for contact-rich dynamics and coordinated bimanual control. The model must handle deformable objects while maintaining sustained force and pose accuracy during the "hold-and-pull" maneuver.

**TubeRack: High-Precision Insertion.** The robot must pick up a test tube and insert it into a rack with a tight tolerance of $< 2\,\text{mm}$ (Fig. 7, bottom). This task demands sub-millimeter spatial precision and strict axial alignment. It provides a rigorous test for the **LAST** framework's ability to bridge high-level intent with low-level precision control under kinematic constraints.

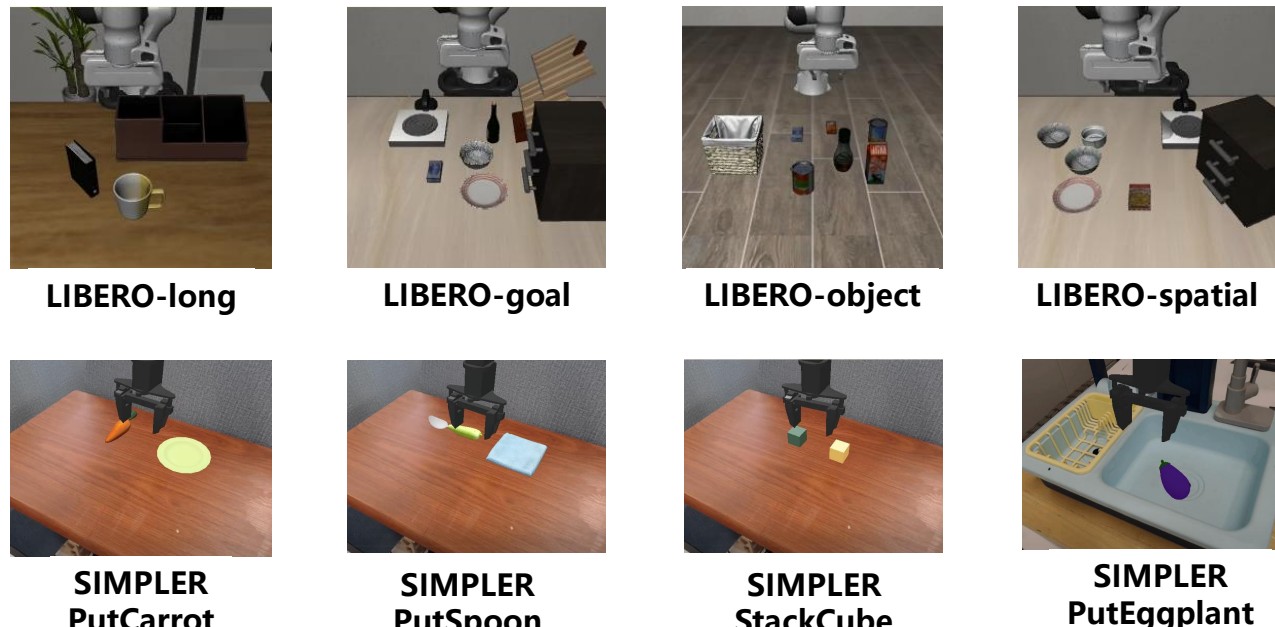

*Figure 8.* **Simulation Benchmark Environments.** (Top) The **LIBERO** suite categorized by long-horizon, goal-conditioned, object-centric, and spatial reasoning tasks. (Bottom) **SimplerEnv** manipulation tasks (Put Carrot, Put Spoon, Stack Block, Put Eggplant) used to evaluate policy robustness under significant domain shifts.

### A.2.2. SIMULATION BENCHMARKS

**LIBERO.** We utilize the LIBERO suite (Liu et al., 2024) to evaluate generalization across spatial, object, goal, and long-horizon categories (Fig. 8, top). This benchmark assesses the policy's ability to adapt to diverse object instances and spatial layouts.

**SimplerEnv.** To test robustness under domain shifts, we evaluate on WidowX robot benchmarks within SimplerEnv (Li et al., 2024a) (Fig. 8, bottom). This evaluates the model's performance under variations in lighting, background, and camera perspectives, following standard evaluation protocols for state-of-the-art VLA comparisons.

### A.3. Model Architecture and Training Details

#### A.3.1. BACKBONE ARCHITECTURE

We adopt **Qwen2.5-VL 3B** as our unified multimodal backbone. The model processes visual inputs through a ViT encoder and interprets them alongside natural language instructions within a 3B-parameter Transformer. To capture fine-grained spatial-temporal features, we utilize an un-pooled vision feature map as the context for the action models. The Transformer backbone produces a sequence of hidden states $\boldsymbol{H} = \{h_1, \ldots, h_L\}$, where each $h_t \in \mathbb{R}^{d_{vlm}}$ encapsulates the multimodal context required for downstream action prediction.

#### A.3.2. HYBRID ACTION HEADS: INTENT AND FLOW

Following the design principles of QwenGR00T (starVLA Contributors, 2025), we equip the model with two parallel action heads that bridge semantic reasoning with physical execution: a **Discrete Intent Head** and a **Continuous DiT Action Flow Head**.

**Discrete Intent Head (Topological Prior).** The discrete head serves as a high-level intent predictor. It consists of a linear projection layer that maps the hidden state $h$ to the logit space of the **LAST** tokenizer's vocabulary $\mathcal{V}$.

**(1) Update Strategy.** During training, we optimize the cross-entropy loss between the predicted logits and the ground-truth

discrete tokens $\bar{q}^{(l)}$ generated by the **LAST** tokenizer:

$$\mathcal{L}_{dis} = -\sum_l \log P(\bar{q}^{(l)} \mid h) = -\sum_l \log \text{Softmax}(\boldsymbol{W}_{dis} h)^{(l)}. \tag{33}$$

**(2) Inference Logic.** During deployment, the head performs autoregressive sampling to determine the optimal action intent $\hat{q}^{(l)} = \arg\max_q P(q \mid h)$. This predicted token provides a topologically grounded prior, effectively selecting the local manifold chart for the subsequent continuous refinement.

**Continuous DiT Action Flow Head (Fine-grained Refinement).** For continuous action generation, we employ a **Diffusion Transformer (DiT-B)** as the action flow head. This head takes the hidden state $h$ as a conditioning signal to perform iterative denoising on the action chunk $\boldsymbol{a} \in \mathbb{R}^{T \times d}$. The DiT head consists of 16 layers with a hidden dimension of 1024, accepting a noisy action $\boldsymbol{a}_k$, a diffusion timestep $k$, and the conditioning feature $h$ injected via cross-attention.

**(1) Update Strategy.** The head is trained to predict the noise $\epsilon$ added to the whitened Lie-algebraic actions using a standard denoising objective:

$$\mathcal{L}_{con} = \mathbb{E}_{\boldsymbol{a}_0, \epsilon \sim \mathcal{N}(0, \mathbf{I}), k} \left[ \|\epsilon - \epsilon_\theta(\boldsymbol{a}_k, k, h)\|^2 \right] \tag{34}$$

where $\boldsymbol{a}_k$ is the noisy action at step $k$. This optimization enables the model to recover the exact local manifold structure within the neighborhood of the selected intent.

**(2) Inference Logic.** We utilize a DDIM scheduler to generate the final continuous action chunk. Starting from pure Gaussian noise $\boldsymbol{a}_K \sim \mathcal{N}(0, \mathbf{I})$, the model performs iterative refinement:

$$\boldsymbol{a}_{k-1} = \text{Denoise}(\boldsymbol{a}_k, k, h; \theta) \tag{35}$$

The resulting continuous trajectory $\boldsymbol{a}_0$ is subsequently mapped back from the $\mathfrak{se}(3)$ Lie algebra to the $SE(3)$ manifold to produce executable physical commands.

**Hybrid Optimization Strategy.** The framework is trained end-to-end using a joint objective that balances discrete intent classification and continuous flow refinement:

$$\mathcal{L}_{vla} = \mathcal{L}_{dis} + \lambda_2 \mathcal{L}_{con}. \tag{36}$$

By setting $\lambda_2 = 10$, the architecture ensures that the backbone's hidden states remain semantically rich while being precisely anchored to the action manifold's requirements.

*Table 7.* Comprehensive Hyperparameters for **LAST**-VLA Training (Qwen2.5-VL-3B + DiT-B).

| Category | Parameters and Values |
|---|---|
| **Architecture** | |
| VLM Backbone | Qwen2.5-VL-3B-Instruct |
| Vision Backbone | Internal ViT (unpooled features) |
| VLM Hidden Dimension | 2048 |
| Action Model Type | DiT-B (16 layers, 1024 dim) |
| Conditioning | Cross-Attention (ada_norm) |
| **VLA Training** | |
| Max Training Steps | 100,000 |
| Warmup Steps | 5,000 (Ratio: 0.05) |
| Base Learning Rate | $1.0 \times 10^{-5}$ (VLM) / $1.0 \times 10^{-4}$ (DiT) |
| LR Scheduler | Cosine with Min LR ($5.0 \times 10^{-7}$) |
| Optimizer | AdamW ($\beta_1 = 0.9, \beta_2 = 0.95, \epsilon = 10^{-8}$) |
| Precision | Mixed Precision (BF16) |
| **Action & Diffusion** | |
| Action Metric | Lie-algebraic $\mathfrak{se}(3)$ (Delta EE) |
| Horizon ($T$) / Dim ($d$) | 16 / 7 |
| Diffusion Steps (Train/Inf) | 4 / 4 (Repeated steps) |
| Noise Schedule | $\beta_\alpha = 1.5, \beta_\beta = 1.0, s = 0.999$ |

Unless otherwise specified, Table 7 reports the default SimplerEnv configuration with $T{=}16$ and $d{=}7$, where the first six dimensions correspond to the $\mathfrak{se}(3)$ delta end-effector motion and the last dimension denotes the gripper command. The LIBERO and real-world settings instead use $T{=}8$ and $T{=}30$ respectively, while keeping $d{=}7$.

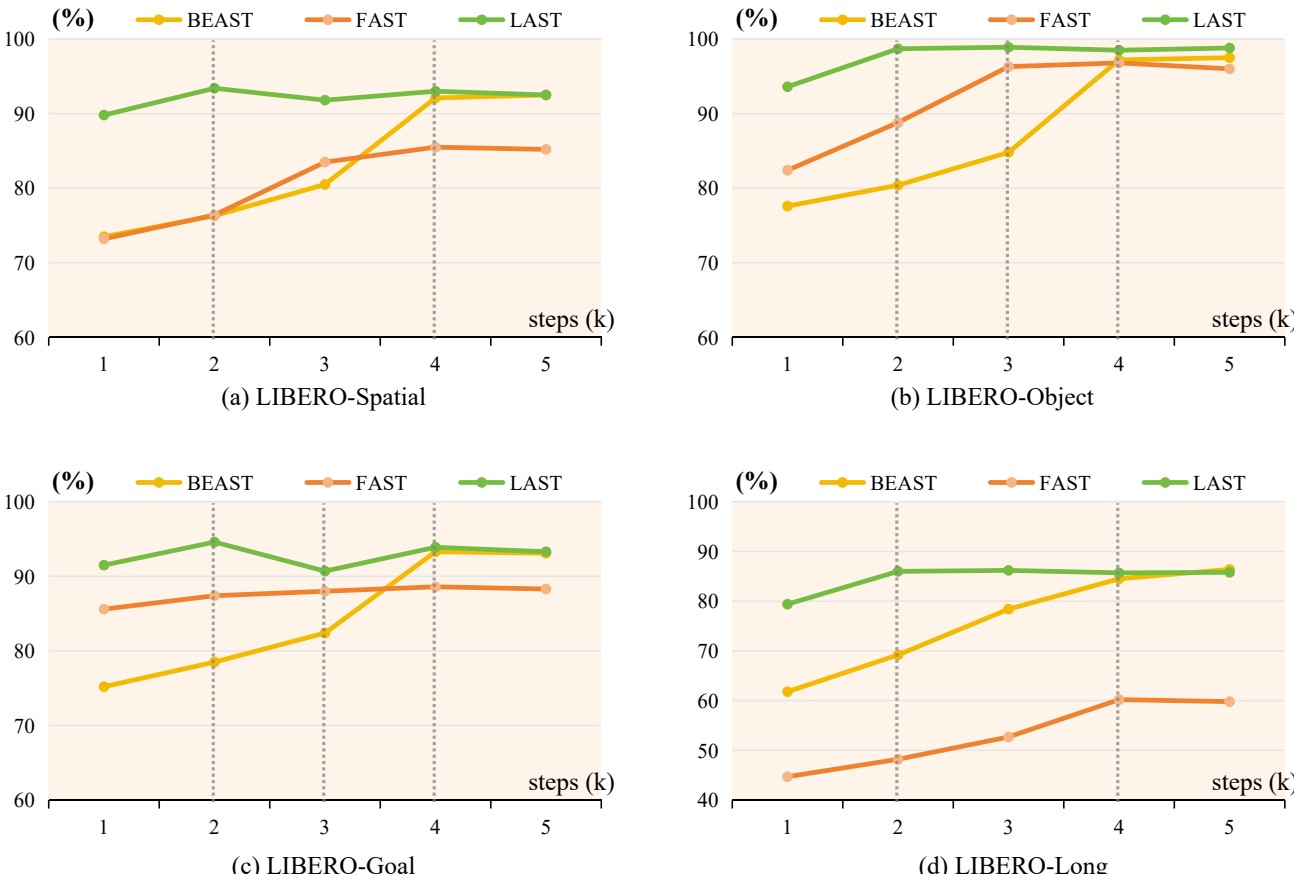

*Figure 9.* **Training Convergence on LIBERO Benchmarks.** Comparison of success rate progression between **LAST** and baselines across four task suites.

### A.4. Supplementary Experimental Results

**Training Efficiency and Manifold Alignment.** We visualize the training convergence across the four LIBERO task suites in Figure 9. The results highlight a critical advantage of the **LAST** framework: **rapid topological alignment**. As observed, **LAST** (green curve) achieves near-optimal performance within the first 1,000 to 2,000 training steps, significantly outpacing geometry-agnostic baselines like FAST and BEAST. This acceleration is particularly pronounced in **LIBERO-Long** (Fig. 9(d)), a horizon-heavy suite requiring sequential consistency. While baselines struggle to learn the long-horizon dependency from scratch via direct regression, **LAST** leverages its pre-learned discrete schemas. Since the intrinsic geometric "shapes" of the actions are already encoded in the tokenizer, the VLA policy need only learn the high-level semantic mapping (*i.e.*, which schema to select) rather than simultaneously learning perception, planning, and low-level control dynamics. This effectively reduces the sample complexity of the downstream task from trajectory regression to intent classification.

#### A.4.1. ABLATION ON COMMITMENT LOSS HYPERPARAMETER $\beta$

The hyperparameter $\beta$ in $\mathcal{L}_{com}$ balances the encoder's output stability and the codebook's learning rate. We evaluate its impact on reconstruction fidelity and codebook health. As shown in Table 8, we observe a clear trade-off:

- **Under-regularization** ($\beta < 0.5$): With low $\beta$ values (0.1, 0.25), the encoder outputs fluctuate loosely around the codebook vectors. While this allows for high freedom, it results in a non-stationary latent target for the VLA, leading to higher reconstruction errors (MAE 0.72e-2).

- **Over-regularization** ($\beta > 0.5$): Conversely, high $\beta$ values (2.0) excessively penalize the encoder, forcing it to collapse to a subset of "safe" codes. This reduces the Codebook Usage (CU) to 81% and increases MAE (0.78e-2) as the model loses the expressivity required to capture fine-grained geometric details.

The default setting of $\beta = 0.5$ achieves the optimal balance, yielding the lowest reconstruction error (0.60e-2) and maximizing codebook utilization (96%), ensuring a rich yet stable vocabulary for the downstream policy.

*Table 8.* Sensitivity analysis of commitment loss weight $\beta$ on Bridge validation set. We observe that $\beta = 0.5$ strikes the best balance between reconstruction quality (MAE) and codebook utilization (CU).

| Weight $\beta$ | 0.1 | 0.25 | **0.5 (Default)** | 1.0 | 2.0 |
|---|---|---|---|---|---|
| MAE ($\downarrow$) | 0.72e-2 | 0.64e-2 | **0.60e-2** | 0.65e-2 | 0.78e-2 |
| CR ($\uparrow$) | 8.0$\times$ | 8.0$\times$ | **8.0$\times$** | 8.0$\times$ | 8.0$\times$ |
| CU ($\uparrow$) | 84% | 92% | **96%** | 93% | 81% |

### A.4.2. SENSITIVITY ANALYSIS OF CODEBOOK SIZE $K$

We evaluate the impact of codebook size $K$ on task performance in the SimplerEnv benchmark. $K$ controls the granularity of topological clustering, determining the level of detail in the action representation.

**Impact of Codebook Size ($K$).** Table 9 shows the effect of codebook size on performance across tasks. With a small codebook ($K = 256$), the model suffers from mode collapse, leading to underfitting and poor performance (*e.g.,* Put Spoon 50.0%, Put Eggplant 66.7%). On the other hand, a large codebook ($K = 4096$) results in sparse clusters, which causes instability in the covariance estimation and whitening process, leading to reduced performance (*e.g.,* 52.1% average SR). The optimal codebook size of $K = 1024$ provides the best balance, yielding the highest success rate (57.3% average SR).

*Table 9.* Success Rate (SR) on SimplerEnv tasks under different codebook sizes $K$.

| Codebook $K$ | Put Spoon | Put Carrot | Stack Block | Put Eggplant | Avg. |
|---|---|---|---|---|---|
| 256 (Small) | 50.0% | 33.3% | 20.8% | 66.7% | 42.7% |
| **1024 (Optimal)** | **62.5%** | **41.7%** | **33.3%** | **91.7%** | **57.3%** |
| 4096 (Large) | 54.2% | 37.5% | 29.2% | 87.5% | 52.1% |

### A.4.3. SENSITIVITY ANALYSIS OF VLA LOSS WEIGHTS $\lambda_1$ AND $\lambda_2$

We evaluate the impact of loss weights on the SimplerEnv benchmark. $\lambda_1$ controls the commitment during tokenizer training, while $\lambda_2$ balances the discrete intent prediction and continuous diffusion refinement during VLA fine-tuning.

**Impact of Tokenizer Regularization ($\lambda_1$).** Table 10 analyzes the impact of the commitment weight $\lambda_1$ during the tokenizer pre-training stage. Since the VLA relies entirely on the learned discrete tokens to form its action plans, the quality of the tokenizer is the bottleneck. Deviation from the optimal $\lambda_1 = 0.5$ causes significant degradation, particularly in high-precision tasks like *Stack Block*. A low $\lambda_1$ results in noisy tokens, while a high $\lambda_1$ restricts the diversity of the action vocabulary.

*Table 10.* Success Rate (SR) on SimplerEnv tasks under different $\lambda_1$ settings (Tokenizer stage).

| $\lambda_1$ Setting | Put Spoon | Put Carrot | Stack Block | Put Eggplant |
|---|---|---|---|---|
| $\lambda_1 = 0.1$ | 29.2% | 33.3% | 12.5% | 62.5% |
| $\lambda_1 = 0.25$ | 45.8% | 41.7% | 20.8% | 79.2% |
| $\lambda_1 = 0.5$ **(Default)** | **62.5%** | 41.7% | **33.3%** | **91.7%** |
| $\lambda_1 = 1.0$ | 58.3% | **47.2%** | 12.5% | 75.0% |
| $\lambda_1 = 2.0$ | 47.2% | 37.5% | 4.2% | 59.2% |

**Balancing Planning and Execution ($\lambda_2$).** Table 11 reveals the necessity of the "Plan-then-Execute" synergy. When the continuous loss is under-weighted ($\lambda_2 = 2.5$), the model correctly identifies intents but fails to ground them physically. Conversely, when the continuous loss dominates ($\lambda_2 = 40.0$), the optimization suffers from "mode averaging," causing catastrophic failure in multi-modal tasks like *Stack Block* (4.2%). Setting $\lambda_2 = 10$ ensures the discrete intent acts as a robust topological anchor while the flow matching head refines local geometry.

*Table 11.* Success Rate (SR) on SimplerEnv tasks under different $\lambda_2$ settings (VLA stage).

| $\lambda_2$ Setting | Put Spoon | Put Carrot | Stack Block | Put Eggplant |
|---|---|---|---|---|
| $\lambda_2 = 2.5$ | 58.3% | 33.3% | 20.8% | 87.5% |
| $\lambda_2 = 5$ | 47.2% | 50.0% | 25.0% | **95.8%** |
| $\lambda_2 = 10$ **(Default)** | **62.5%** | 41.7% | **33.3%** | 91.7% |
| $\lambda_2 = 20$ | 33.3% | 45.8% | 16.7% | 79.2% |
| $\lambda_2 = 40$ | 50.0% | **54.2%** | 4.2% | 70.8% |

## A.5. B-Spline Algorithm Details

**B-spline basis.** A univariate B-spline of degree $p$ is defined over a nondecreasing knot vector $\mathcal{U} = \{u_0, \ldots, u_M\}$ with $M = T_c + p$ for $T_c$ control points. The $p$-degree basis functions $\{N_{i,p}(u)\}_{i=0}^{T_c-1}$ are given by the Cox–de Boor recursion:

$$N_{i,0}(u) = \begin{cases} 1, & u_i \leq u < u_{i+1}, \\ 0, & \text{otherwise}, \end{cases} \tag{37}$$

$$N_{i,p}(u) = \frac{u - u_i}{u_{i+p} - u_i} N_{i,p-1}(u) + \frac{u_{i+p+1} - u}{u_{i+p+1} - u_{i+1}} N_{i+1,p-1}(u), \tag{38}$$

with the convention $0/0 := 0$. We employ *uniform clamped* knots (first and last knots repeated $p+1$ times), which ensure interpolation at the endpoints and stable evaluation.

**Trajectory model.** A B-spline trajectory of degree $p$, parameterized over $u \in [0, 1]$ and defined by control points $\boldsymbol{c} = [c_0, \ldots, c_{T_c-1}]^\top$, is given by

$$y(u) = \sum_{i=0}^{T_c-1} c_i N_{i,p}(u), \qquad u \in [0, 1]. \tag{39}$$

Given a normalized action sequence $a_{1:T} = [a_1, \ldots, a_T]$ with length $T$, the objective is to construct a B-spline trajectory $y(u)$ that approximates the action sequence. A linear transformation maps the action timestep to the parametric space of the spline trajectory, where the sampled grid is defined as

$$u_\tau = \frac{\tau - 1}{T - 1}, \qquad \tau = 1, \ldots, T. \tag{40}$$

**Design matrix and least squares.** To make the B-spline trajectory on the sampled grid $y(u)_{1:T}$ closely matches the action sequence $a_{1:T}$, the control points are obtained by minimizing the least-squares error

$$\boldsymbol{c}^\star = \arg\min_{\boldsymbol{c}} \left\| y(u)_{1:T} - a_{1:T} \right\|_2^2. \tag{41}$$

We further formalize the precomputed B-spline basis functions of $y(u)$ into the B-spline design matrix $\boldsymbol{\Phi} \in \mathbb{R}^{T \times T_c}$, where

$$\Phi_{\tau,i} = N_{i,p}(u_\tau), \qquad \tau = 1, \ldots, T, \; i = 0, \ldots, T_c - 1. \tag{42}$$

Under this formulation, estimating the B-spline trajectory parameters reduces to a standard least-squares optimization problem:

$$\boldsymbol{c}^\star = \arg\min_{\boldsymbol{c}} \left\| \boldsymbol{\Phi}\boldsymbol{c} - a_{1:T} \right\|_2^2, \quad \text{(unweighted)} \tag{43}$$

$$\boldsymbol{c}^\star = \arg\min_{\boldsymbol{c}} \left\| \boldsymbol{W}^{1/2}(\boldsymbol{\Phi}\boldsymbol{c} - a_{1:T}) \right\|_2^2 \quad \text{(weighted)}, \tag{44}$$

where $\boldsymbol{W} \succeq 0$ can emphasize specific timestamps (*e.g.,* contacts).

**Ridge regularization.** To improve numerical stability and avoid overfitting when the number of control points $T_c$ is large or the samples are noisy, we introduce an $\ell_2$ penalty, parameterized by a regularization coefficient $\lambda$ ($\lambda > 0$):

$$\boldsymbol{c}^\star = \arg\min_{\boldsymbol{c}} \left\| \boldsymbol{\Phi}\boldsymbol{c} - a_{1:T} \right\|_2^2 + \lambda \|\boldsymbol{c}\|_2^2 = (\boldsymbol{\Phi}^\top \boldsymbol{\Phi} + \lambda \boldsymbol{I})^{-1} \boldsymbol{\Phi}^\top a_{1:T}. \tag{45}$$

In practice we precompute $\boldsymbol{\Phi}$ from equation 42 and solve equation 45 via a Cholesky factorization of $(\boldsymbol{\Phi}^\top \boldsymbol{\Phi} + \lambda \boldsymbol{I})$. This batched procedure incurs only a minor computational overhead, typically on the order of a few milliseconds.

**From 1-DoF to multi-DoF.** For a $d$-DoF action sequence $\boldsymbol{a}_{1:T} \in \mathbb{R}^{T \times d}$, we fit each DoF independently using the same time grid $u_{1:T}$ and design matrix $\boldsymbol{\Phi}$:

$$\boldsymbol{c}_j^\star = (\boldsymbol{\Phi}^\top \boldsymbol{\Phi} + \lambda \boldsymbol{I})^{-1} \boldsymbol{\Phi}^\top a_{1:T}^{(j)}, \qquad j = 1, \ldots, d. \tag{46}$$

Stacking $\{\boldsymbol{c}_j^\star\}_{j=1}^d$ yields the control-point matrix

$$\boldsymbol{C} = \begin{bmatrix} (\boldsymbol{c}_1^\star)^\top \\ \vdots \\ (\boldsymbol{c}_d^\star)^\top \end{bmatrix} \in \mathbb{R}^{d \times T_c}, \tag{47}$$

which serves as the fixed-length representation. Reconstruction at any $u$ follows from equation 39 using the corresponding row of $\boldsymbol{\Phi}$.

**Choice of degree, knots, and control points.** We use clamped uniform knot sequences, with spline degree set to $p{=}4$ for the smooth, high-fidelity representation of position and rotation trajectories, and $p{=}0$ for the near piecewise-constant representation of gripper signals. The number of control points $T_c$ balances reconstruction accuracy against representation compactness. In practice, we choose $T_c \ll T$ to align variable horizons into a common fixed length while retaining millimeter-level reconstruction.

## A.6. Evaluation Metrics

To comprehensively evaluate the **LAST** framework and its impact on VLA training, we define a multi-faceted set of metrics. These metrics assess the model from the perspectives of physical fidelity, structural efficiency, and closed-loop performance.

**Mean Absolute Error (MAE):** This metric quantifies the **physical reconstruction fidelity** between the ground-truth expert trajectories and the reconstructed actions.

$$\text{MAE} = \frac{1}{T \cdot d} \sum_{t=1}^{T} \sum_{i=1}^{d} |a_{t,i} - \hat{a}_{t,i}| \tag{48}$$

where $T$ is the trajectory horizon and $d$ is the action dimension. In robotic control, MAE serves as an open-loop proxy for the tokenizer's ability to preserve high-frequency motion details. A low MAE ensures that the discretization process does not filter out critical control nuances, such as precise gripper orientations or contact forces.

**Compression Ratio (CR):** CR measures the **bottleneck efficiency** of the action representation, defined as the ratio between raw data size and the resulting token sequence length.

$$\text{CR} = \frac{T \times d}{N_{\text{tokens}}} \tag{49}$$

where $N_{\text{tokens}}$ is the number of discrete tokens used to represent a trajectory. For Transformer-based VLA models, CR is a critical determinant of computational overhead, as the attention complexity scales quadratically with sequence length. **LAST** aims to achieve a high CR by leveraging Lie-algebraic priors to represent complex $SE(3)$ motions with minimal discrete intents.

**Codebook Utilization (CU):** CU evaluates the **statistical efficiency** of the vector quantization (VQ) process by measuring the proportion of active entries in the codebook $\mathcal{C}$ during validation.

$$\text{CU} = \frac{1}{|\mathcal{C}|} \sum_{e \in \mathcal{C}} \mathbb{I}\left[\text{count}(e) > 0\right] \tag{50}$$

A low CU often signifies "representation collapse" or the presence of "dead codes," typically caused by the anisotropic nature of the raw action space. In our framework, high CU values validate the effectiveness of *covariance-aware whitening*, which reshapes the action clusters into an isotropic distribution that matches the Euclidean geometry of the VQ codebook.

**Success Rate (SR):** SR is the primary metric for **closed-loop policy performance**, representing the probability of a task being fully completed during deployment.

$$\text{SR} = \frac{1}{N} \sum_{i=1}^{N} \text{TaskCompleted}(i) \tag{51}$$

Unlike open-loop metrics, SR evaluates the model's robustness to compounding errors and its ability to generalize to novel object poses and environmental perturbations. It reflects how effectively the "Topological Complexity Collapse" (Lemma A.3) facilitates the mapping from semantic reasoning to precise physical execution.

