# OpenReview forum: "LAST: Bridging Vision-Language and Action Manifolds via Gromov-Wasserstein Alignment"
_ICML.cc/2026/Conference — ICML 2026 regular_

### Official Review · Reviewer_AuSJ · 2026-02-28

**Soundness:** 3
**Presentation:** 4
**Significance:** 4
**Originality:** 3
**Overall Recommendation:** 5
**Confidence:** 3

**Summary:**

The authors propose a method to bridge the gap between VLM semantic and robot action spaces using algebraic techniques that can linearize and normalize complex robotic movement trajectories. They provide reasoning and mathematic evidence on why this alignment allows VLMs to directly map correct physical motions. Their resulting overall framework is evaluated across two simulated environments and three real world tasks.

**Compliance With Llm Reviewing Policy:**

Affirmed.

**Key Questions For Authors:**

N/A

**Strengths And Weaknesses:**

**Strengths**

1. Rigorous mathematical motivation and foundation
2. Strong reconstruction ability is promising
3. Through experimentation and strong results on established (simulated) benchmarks
4. Complex real world tasks like zipping a bag


**Weaknesses**

1. Complex Data Preprocessing - lacks discussion of compute / latency of these.
2. Two Stage Training process required (which can be less favorable than end-to-end fine-tuned VLAs).
3. Details on overall framework inference latency missing.

---

> ### Author Rebuttal · Authors · 2026-03-31
>
> ## W1:
>
> The preprocessing in LAST is **not** a large-scale optimization that must be repeatedly solved at every control step. Instead, it is better viewed as a lightweight offline preprocessing module, **with computational cost below 1% of VLA fine-tuning** (specifically, 0.85% on LIBERO and 0.06% on SimplerEnv). More importantly, the convergence curves in Figure 9 already show that LAST typically reaches near-optimal performance within about 1k–2k steps on the four LIBERO task suites, clearly faster than FAST and BEAST. This indicates that, although LAST introduces a structured tokenizer and manifold rectification, it does not slow down optimization; instead, by encoding action geometry through discrete schemas, it shifts downstream VLA learning from direct trajectory regression to more stable high-level intent selection, resulting in **faster convergence**. We summarize the latency measurements in **W3**.
>
>
> ## W2:
>
> We would like to clarify the training pipeline of LAST first:
>
> (1) **Tokenizer pretraining (offline, reusable):** we pretrain an adaptive action tokenizer once to build a shared discrete action space. This is a common prerequisite in many modern VLA pipelines (e.g., pi0.5[2]) and is amortized across datasets, rather than retrained separately for each benchmark.
>
> (2) **VLA dual-objective learning (single fine-tuning stage):** the VLA is then fine-tuned end-to-end under a matched budget with both discrete supervision $L_{\text{dis}}$ and continuous flow matching. This is a single VLA training stage, not an additional per-dataset two-stage VLA procedure.
>
> Hence, LAST should be viewed as adding a reusable tokenizer pretraining component plus a single-stage end-to-end VLA fine-tuning objective, rather than introducing an unfavorable multi-stage VLA optimization. Meanwhile, as shown in Fig. 3, tokenizer pretraining also improves fitting speed during training.
>
> Empirically, under the same backbone, data, and training budget, replacing the dual objective with direct flow-only learning degrades performance:
>
> | Method | LIBERO Avg.  | SimplerEnv Avg.  | PlaceObj  | ZipSeal  | TubeRack  |
> | -------------------- | ------------: | ----------------: | ---------: | --------: | ---------: |
> | Direct Flow Matching | 95.1 | 51.6 | 45 | 50 | 35 |
> | LAST | 95.8 | 57.3 | 73 | 57 | 48 |
>
> As shown above, direct end-to-end flow matching performs worse than LAST on LIBERO, SimplerEnv, and all three real-world tasks, with especially large gaps on real-robot settings. This indicates that the gain comes from the dual-objective inductive bias during one-stage VLA fine-tuning, not from imposing an extra per-dataset multi-stage training burden.
>
>
> ## W3:
>
> We have added a dedicated Latency / Efficiency table in the revision, reporting single-step inference latency (per action chunk), throughput, and inference TFLOPs/step, with comparisons to FAST and BEAST.
>
> | Method | Single-step Latency (ms) ↓ | Throughput (Hz)  | inference TFLOPs / step ↓ |
> | ------ | -------------------------: | ----------------: | ------------------------: |
> | FAST | 62.58 | 255.67 | 2.038278 |
> | BEAST | 63.11 | 253.53 | 2.045360 |
> | LAST | 63.82 | 250.71 | 2.047565 |
>
> Compared with FAST, LAST adds only +1.24 ms latency per step (63.82 vs 62.58, about +1.98%) and -4.96 Hz throughput; compared with BEAST, the latency gap is +0.71 ms. This indicates that the overall framework inference latency of LAST is comparable to strong baselines. Overall, the main additional cost of LAST lies in offline structured modeling, rather than expensive online control-time computation, while the existing convergence results show that this extra structure leads to **faster and more stable optimization**.

---

> > ### Author Rebuttal · Reviewer_AuSJ · 2026-04-02
> >
> > Concerns resolved, retaining accept rating.

---

> > > ### Author Response · Authors · 2026-04-07
> > >
> > > We sincerely thank you for your thoughtful review and for highlighting concerns about efficiency, training design, and latency.
> > >
> > > Your practical perspective is greatly appreciated, and we are thankful for the time and care you devoted to assessing our work. Thank you again for your valuable feedback and consideration!

---

### Official Review · Reviewer_cbw4 · 2026-03-09

**Soundness:** 3
**Presentation:** 3
**Significance:** 3
**Originality:** 3
**Overall Recommendation:** 4
**Confidence:** 3

**Summary:**

This paper take VLA learning as a Gromov-Wasserstein (GW) alignment problem and propose the Lie-algebraic Action Space Tokenizer (LAST). LAST transforms action trajectories through a two-stage process: (1) Global Topological Linearization, which maps non-additive $SE(3)$ actions into the $\mathfrak{se}(3)$ tangent space and abstracts them using B-splines ; and (2) Local Metric Discretization, which applies covariance-aware whitening to rectify local anisotropy, establishing an isotropic Euclidean metric compatible with VL embeddings. The experiments show a great performance.

**Compliance With Llm Reviewing Policy:**

Affirmed.

**Final Justification:**

The author solved my concerns during rebuttal, I will keep my score.

**Key Questions For Authors:**

Please refer to the weakness.

**Limitations:**

Please refer to the weakness.

**Strengths And Weaknesses:**

## Strength
**Soundness** The paper is technically robust and mathematically rigorous. The authors correctly identify the issue of Euclidean non-additivity in standard tokenizers (such as RVQ or binning) and provide an elegant solution by mapping to the Lie algebra $\mathfrak{se}(3)$. Furthermore, the introduction of the covariance-aware whitening transformation, $\mathcal{W}_{soft}$, effectively addresses statistical mismatch by preconditioning the manifold geometry to ensure isotropic gradient steps.

**Presentation** The paper is well-written and easy to follow.

**Novelty** The author(s) apply Lie algebra and B-splines technique to action tokenization is novel.

## Weakness

1. While the results on LIBERO and SIMPLER provide a baseline, they are somewhat insufficient to fully validate the proposed method.
2. The method relies on maintaining a running estimate of local covariance $\Sigma_k$ during the tokenizer's training phase. The paper does not thoroughly analyze the stability. How sensitive of this hyper-parameter to affect the training process and the final results?

---

> ### Author Rebuttal · Authors · 2026-03-31
>
> ## W1:
>
> Our current paper provides a three-level evidence chain, rather than relying on a single simulation benchmark.
>
> 1. At the tokenizer level, Table 1 in the main paper shows that LAST achieves both lower reconstruction error and better structural efficiency, clearly outperforming standard RVQVAE.
> 2. At the simulation level, LAST-Continuous reaches 95.8% on LIBERO and 57.3% on SimplerEnv.
> 3. At the real-world level, Sec. 4.2–4.4 already includes three real robot tasks, PlaceObj, ZipSeal, and TubeRack, where LAST achieves 73% / 57% / 48%, significantly outperforming Pi-FAST and BEAST.
>
> To further strengthen both comparison fairness and result stability, we present more direct pieces of evidence on Calvin:
>
> 1. a matched-setting baseline experiment, using the same backbone, Qwen2.5, the same StarVLA framework, and the same training budget, batch size, action horizon, and two-stage training recipe, while only changing the action representation / tokenizer design;
> 2. a multi-seed reproducibility experiment, reporting mean ± std over 3 random seeds.
>
> | Method | ABC$\rightarrow$D | ABCD$\rightarrow$D | Avg.  |
> |---|---:|---:|---:|
> | Qwen2.5-LAST | 4.35±0.08 | 4.51±0.03 | 4.43±0.06 |
> | Qwen2.5-FAST | 4.10±0.18 | 4.25±0.05 | 4.18±0.12 |
> | Qwen2.5-BEAST | 3.88±0.02 | 4.24±0.16 | 4.06±0.09 |
>
> These additional results show that:
>
> 1. Under a fully matched setting, LAST still remains the best-performing method on Calvin, indicating that its gains come from the structured action interface built by Lie-algebraic linearization + covariance-aware whitening.
> 2. Across 3 random seeds, LAST exhibits consistently small variance, suggesting that its improvements are not due to a lucky initialization, but are stable and reproducible.
>
> Therefore, beyond LIBERO/SimplerEnv, the added real-world and Calvin matched-setting results provide out-of-benchmark evidence for both generalization and robustness.
>
> ## W2:
>
> To address the concern that cross-dataset instability of $\Sigma_k$ could undermine whitening reliability, we clarify the role of $\Sigma_k$ in LAST. In LAST, $\Sigma_k$ is not a global parameter expected to remain fixed across datasets; rather, it is a local statistical normalizer defined under a fixed local chart. Therefore, we have added an experiment to explore whether $\Sigma_k$ can be estimated stably under the same charting scheme and whether it exhibits severe drift under reduced data.
>
> To this end, we add a targeted stability experiment. Under the same coarse schema selection, we independently estimate $\Sigma_k$ from 10% subsets and the full training set on LIBERO, SimplerEnv, and the real-world dataset, and compare their similarity using:
>
> - Frobenius Similarity, which measures the agreement in overall covariance structure
> - Principal-direction Consistency, which measures the consistency of dominant principal directions
>
> | Dataset | Subset Ratio | Frobenius Similarity  | Principal-direction Consistency  |
> | ------- | -----------: | ---------------------: | --------------------------------: |
> | LIBERO | 10% vs 100% | 0.951 | 0.916 |
> | SimplerEnv | 10% vs 100% | 0.854 | 0.842 |
> | Real-world | 10% vs 100% | 0.889 | 0.905 |
>
> These results suggest that:
>
> 1. A relatively stable $\Sigma_k$ can already be estimated without requiring the full dataset; even with only 10% of the data, the estimated local covariance remains highly consistent with the full-data estimate.
>
> 2. Although $\Sigma_k$ is dataset-dependent, it does not exhibit severe instability or unusable drift; in other words, LAST relies on local statistical structure that can be estimated reliably, rather than on a fragile covariance parameter.
>
> Together with Supplementary Tables 8–11 (sensitivity to $K$, $\beta$, $\lambda_1$, and $\lambda_2$), these results indicate that LAST operates in a broad and interpretable stability region, rather than depending on a fragile covariance estimate.

---

> > ### Author Rebuttal · Reviewer_cbw4 · 2026-04-02
> >
> > Thanks for the author's response. My concerns have been adequately addressed.

---

> > > ### Author Response · Authors · 2026-04-07
> > >
> > > Thank you for your thoughtful review! We especially appreciate your comments on validation scope and covariance stability.
> > >
> > > Your feedback has been very valuable to us, and we fully respect your current assessment. We hope our rebuttal helps resolve the main concerns, and we would be very grateful for any further consideration of the score during the discussion process.

---

### Official Review · Reviewer_sDzt · 2026-03-12

**Soundness:** 3
**Presentation:** 3
**Significance:** 3
**Originality:** 3
**Overall Recommendation:** 4
**Confidence:** 2

**Summary:**

The paper studies action tokenization for Vision-Language-Action (VLA) models through embedding geometry. The semantic space of vision-language is topologically linear and isotropic, whereas the physical manifold of robotic action is non-Euclidean and anisotropic tied to SE(3) motion. This mismatch makes standard Euclidean regression or geometry-agnostic tokenization poorly suited for cross-modal VL-A alignment.

By reconstructing the action space, LAST (Lie-algebraic Action Space Tokenizer) establishes a more consistent metric alignment between the VL and Action modalities. LAST linearizes the action manifold via Lie-algebraic mapping, converting trajectories into a fixed-length, physically additive representation with B-spline control points. Finally, they discretize the representation into schemas and local covariance-aware whitening residuals, establishing a mathematical isomorphism with the isotropic Euclidean metric. The tokenizer is then integrated into a VLA pipeline with both discrete token supervision and a continuous refinement head.

Empirically, the author reports reconstruction/compression trade-offs than several tokenization baselines (Binning, FAST, RVQVAE), along with improved results on LIBERO, SimplerEnv, and three real-world manipulation tasks (PlaceObj, ZipSeal, and TubeRack).

**Compliance With Llm Reviewing Policy:**

Affirmed.

**Final Justification:**

The authors have adequately addressed the reviewers’ comments and clarified several points in the revised manuscript. The changes improve readability and presentation, but they do not substantially alter the work's overall impact. Therefore, my evaluation and score remain unchanged.

**Key Questions For Authors:**

1. The paper frames LAST as solving or approximating a **Gromov-Wasserstein alignment** problem, but the training procedure appears not to optimize an explicit GW objective. Can the authors clarify exactly in what sense LAST is a GW method rather than a **geometry-inspired tokenizer?**
2. How **sensitive** is LAST to the local charting and whitening assumptions in practice?
3. Can the authors provide **statistical variance** over multiple random seed runs? This will strengthen the method’s reliability.
4. A clearer discussion of what is proven, what is assumed, and what is only an intuition would improve the paper.
5. The paper also needs a stronger discussion of comparison fairness and reproducibility details.

**Limitations:**

Partially. The discussion of limitations is too weak. The impact statement is very brief and does not seriously discuss realistic limitations or risks.  The manuscript would benefit from a clearer discussion of assumptions. The supplementary material includes sensitivity analyses for several hyperparameters (e.g., codebook size, commitment loss, and loss balancing weights), which provides some insight into the robustness of the approach.

**Strengths And Weaknesses:**

**Soundness:** This paper has a clear core idea: better action tokenization may require respecting the geometry of robot action spaces rather than forcing them into a flat Euclidean representation. The methodology is built around this idea. The design is reasonably complete: tokenizer pre-training (a global step for manifold linearization and a local step for whitening and quantization) and VLA Dual-object training (Predict discrete action tokens with discrete head (topological alignment) and continuous head (Metric refinement)).

The experiments are extensive and evaluate both tokenizer quality and downstream policy learning. The ablations are useful evaluating effect of se(3) manifold alignment, tokenizer training objectives, continuous head architecture and discrete supervision. However, several theoretical statements rely on fairly strong assumptions, such as Gaussian-mixture structure, local chart validity, and whitening producing near-isotropic local behavior which makes method feel technically plausible and well-engineered. Some therotical claims are stronger than evidence/results support.


**Weakness:**
1.  “GW alignment” framing feels more conceptual than algorithmically realized. I did not see an explicit Gromov-Wasserstein objective, transport plan optimization, or a direct empirical study showing that GW-style structural alignment itself is what drives the gain.
2. Are all baseline comparisons performed under the same backbone, training budget, data, and implementation pipeline? (PS: VLAs are very sensitive to training details)

**Presentation:** The paper is structured well. The problem statement, method, and experimental results are easy to follow. Figures 1 and 2 are helpful, and the narrative from geometric mismatch to tokenizer design is clear.

**Suggestions:**
1. The paper is theoretically sound but some notation and wording could also be tightened.
2. A clearer discussion of what is proven, what is assumed, and what is only an intuition would improve the paper.
3. The paper also needs a stronger discussion of comparison fairness and reproducibility details.


**Significance:** The topic of action tokenization is central for scalable VLA systems. The model could be useful both practically and conceptually, especially for long-horizon manipulation or settings where action geometry matters. I do think the paper’s impact is somewhat limited by the fact that the strongest claim is about a tokenizer/interface rather than a fully new VLA paradigm, but that **is still a worthwhile contribution.**

**Originality:** The paper combines known principles in a thoughtful engineered way. Lie-algebraic representations, spline parameterizations, vector quantization, and whitening are all familiar ideas individually. The **novel part is the specific combination**, the framing of tokenization as a geometric alignment problem, and the way local whitening is used to make residual quantization better. I think the **novelty is stronger in system design and framing than in deep theoretical advance**.

---

> ### Author Rebuttal · Authors · 2026-03-31
>
> ## KQ1&W1:
> We clarify that LAST is **not** an explicit GW/OT solver; it is a **GW-motivated structural surrogate**.
>
> Instead of optimizing a transport plan, LAST uses two tractable components: (i) schema-level discrete decomposition for coarse relational topology alignment, and (ii) chart-wise local refinement with covariance-aware whitening to reduce action-space anisotropy and improve local metric matching. This follows the GW principle of aligning intra-space relational structure rather than pointwise coordinates.
>
> To explain why we do not use explicit GW optimization, we add an explicit relation loss $\mathcal{L}_{GW}=\|R^{vl}-R^{act}\|_F^2$ in LMD (where $R^{vl}$/$R^{act}$ are batch-wise relation matrices of VL states/action latents):
>
> |Method|LIBERO|SimplerEnv|
> |-|-:|-:|
> |LAST|95.8|57.3|
> |LAST with $\mathcal{L}_{GW}$|95.3|54.1|
>
> As shown in Table 1, explicitly adding $\mathcal{L}_{GW}$ does not improve performance. A likely reason is that explicit GW-style supervision (akin to contrastive-style relational modeling) is more scaling-sensitive, while VLA fine-tuning under limited budgets is relatively insensitive to it. Thus, in our setting, GW is better used as a modeling principle for structural decomposition than as a directly optimized transport objective.
>
> ## KQ2:
> To assess sensitivity to local charting and whitening, we add a real-world sweep over chart size $K$. Each cell reports **with whitening / without whitening** performance:
>
> |Codebook $K$|PlaceObj|ZipSeal|TubeRack|
> |-|-:|-:|-:|
> |256|75/53|50/32|42/26|
> |512|70/55|55/40|50/30|
> |1024(Default)|73/65|57/40|48/42|
> |2048|68/50|58/36|46/38|
> |4096|72/58|53/38|48/42|
>
> Overall, LAST remains stable across a broad range of $K$ (roughly 512-4096), while whitening consistently improves performance, indicating a robust rather than brittle design.
>
> ## KQ3:
> To address random-seed variance, we report mean±std over 3 seeds. All methods use the same seed set, data split, and checkpoint-selection protocol.
> |Method|LIBERO|SimplerEnv|Real-World|
> |-|-:|-:|-:|
> |LAST|95.8±0.1|57.3±2.4|59.3±3.6|
> |FAST|94.1±0.3|45.8±4.6|31.3±6.0|
> |BEAST|92.9±0.7|40.6±6.9|43.7±1.6|
>
> The results support that LAST's gains are reliable.
>
> ## KQ4:
> To avoid conflating assumptions, derivations, and intuition, we separate the theory into:
> 1. **Assumptions**:
>    - The conditional action distribution has a multimodal structure;
>    - The action manifold can be approximated by local charts;
>    - Residuals within each chart can be modeled by local statistics ($\mu_k,\Sigma_k$).
> 2. **Derive under assumptions**:
>    - Direct Euclidean regression yields off-manifold mean solutions on multimodal, non-convex action manifolds in Sec.A.1.1;
>    - Local covariance-aware whitening improves the local metric condition, rather than “creating semantic alignment” by itself in Sec.A.1.2;
>    - A hybrid structure of discrete schema + local refinement is more favorable for high-dimensional action spaces in terms of complexity in Sec.A.1.3;
>    - Lie-algebraic linearization + B-spline abstraction provides an approximately additive representation only within local charts, with controlled error in Sec.A.1.4.
> 3. **Intuition**: discrete schemas first identify the correct high-level action mode, and local whitening then models fine-grained continuous control within that mode. Together, this makes learning more stable under multimodality.
>
>
> ## KQ5 & W2:
> For fairness, all reproduced baselines follow default recommended hyperparameters and use exactly matched training data, budget, and evaluation protocol. We further provide a matched-setting baseline using the same backbone (Qwen2.5) and exactly the same hyperparameters in the StarVLA[1] framework.
>
> For reproducibility, Sec. 4.1 reports key LAST training configurations, and the supplement provides full architecture, optimizer, LR, diffusion, and precision settings. **We will also release VLA fine-tuning and tokenizer pretraining scripts for all main-table results upon acceptance.**
>
> |Method|Spatial|Object|Goal|Long|Avg.|
> |-|-:|-:|-:|-:|-:|
> |LAST|98.6±0.2|97.2±0.1|96.7±0.1|91.5±0.3|96.0±0.2|
> |FAST|97.1±0.3|96.9±0.1|94.8±0.2|89.2±0.1|94.5±0.2|
> |BEAST|94.3±0.3|96.1±0.5|93.9±0.1|88.4±1.1|93.2±0.5|
>
> |Method|PutSpoon|PutCarrot|StackBlock|PutEggplant|Avg.|
> |-|-:|-:|-:|-:|-:|
> |LAST|65.3±2.4|43.1±4.2|29.2±0.0|93.1±2.4|57.7±2.4|
> |FAST|45.8±4.2|37.5±4.2|18.5±7.9|79.2±7.2|45.3±5.9|
> |BEAST|40.3±2.4|33.3±8.4|16.7±4.2|77.8±8.7|42.2±5.9|
>
> In the revision, we will strengthen the Limitations and Impact sections by explicitly discussing practical assumptions, realistic risks, and key hyperparameter sensitivity findings to better contextualize robustness.
>
> [1] starVLA Contributors. StarVLA: A Lego-like Codebase for Vision-Language-Action Model Developing. GitHub repository, 2025. DOI:10.5281/zenodo.18264214.
>
> [2] Physical Intelligence Team. pi0.5: A Vision-Language-Action model for robotics. Technical report, 2025.

---

> > ### Author Rebuttal · Reviewer_sDzt · 2026-04-02
> >
> > My concerns have been adequately addressed by the author's responses. Retaining my score.

---

> > > ### Author Response · Authors · 2026-04-07
> > >
> > > We greatly appreciate your thoughtful review and your careful attention to GW alignment and baseline fairness！
> > >
> > > We fully respect your current assessment and are thankful for the time and effort you devoted to evaluating our work. If our rebuttal and clarifications encourage any further consideration during the discussion process, we would be sincerely grateful.

---

### Decision · Program_Chairs · 2026-04-30

**Decision:**

Accept (regular)

**Comment:**

This paper introduces a novel action tokenizer for VLA that linearizes SE(3) trajectories, providing a more effective representation for distance minimization objectives. The reviews are overall positive: reviewers found the method technically solid and the results strong. Major concerns are that (1) the GW alignment claim is more conceptual than explicit, and (2) the robustness of discretization needs better justification. In the rebuttals, the authors acknowledged that LAST is GW-motivated; new evidence about multi-seed, stability, and matched settings is added. All reviewers indicated that the concerns are largely resolved. Final ratings are all positive.

The AC's main remaining concern is the missing matched-setting baseline on the main benchmarks. The paper does not compare against a StarVLA/GR00T-style baseline on LIBERO or SimplerEnv under the same framework and backbone, and the rebuttal only adds Calvin results. Thus, it remains unclear how much of the gain comes from LAST itself. Given the otherwise positive review profile, I recommend Weak Accept.